# Dynamic Transition and Convergence Trend of the Innovation Efficiency among Companies Listed on the Growth Enterprise Market in the Yangtze River Economic Belt—Empirical Analysis Based on DEA—Malmquist Model

**Yanqi Han \*, Minghui Hua, Malan Huang, Jin Li and Shirui Wang**

School of Business, Hubei University, Wuhan 430062, China; 202121112010687@stu.hubu.edu.cn (M.H.); huangml2019@hubu.edu.cn (M.H.); 201911111210354@stu.hubu.edu.cn (J.L.); 201822211220112@stu.hubu.edu.cn (S.W.)

\* Correspondence: hanyanqi321@hubu.edu.cn; Tel.: +86-1582-7091-669

**Abstract:** Background: The Yangtze River Economic Belt (YREB) occupies an important economic position in China and has great research value. Methods: Based on the panel data of 142 GEM-listed companies in the YREB from 2015 to 2019, using the DEA Malmquist index, $\sigma$-convergence and $\beta$-convergence models, this study empirically analyzes the dynamic change and convergence trend of the innovation efficiency of these companies. Results: The number of these companies increased significantly but the innovation efficiency of them has not reached the optimal level. From a static point of view, companies in the middle reaches of the Yangtze River have the highest innovation efficiency, while from the dynamic point of view, the Yangtze River Delta region has the highest innovation efficiency. Moreover, most companies have an agglomeration effect, and there is a big gap in innovation efficiency. There is no $\sigma$-convergence trend in the YREB and its sub-regions, but there is an obvious $\beta$-convergence trend. Conclusions: The innovation efficiency of these companies has a lot of room for improvement. There is industry heterogeneity, and exogenous factors have different effects on the improvement of innovation efficiency in different regions owing to the differences in geographical location, economic development level, and other factors.

**Keywords:** Yangtze River; economic belt; GEM-listed companies; innovation efficiency; dynamic transition; convergence trend

## 1. Introduction

The Yangtze River Economic Belt (YREB) traverses the three major East West Regions of China, with a geographical area of approximately 2,050,000 km². It covers 21% of the total land area in China, with a population and economic sum exceeding 40%. The YREB has a significant ecological position, and great comprehensive strength. In terms of development potential, it is the leading innovation driver belt for the transformation and development of China, providing the inner river economic belt with global influence, and coordinated development of East and West interactions and cooperation. The outline of the development plan for the YREB proposes reinforcement of the transformation and upgrade of innovation-driven industries in the YREB. Most importantly, it proposes to enhance the autonomous innovation ability of the urban cluster of the YREB and related industries, fully implement technological innovation engineering, integrate the advantages of innovative resources, and create an industrial technology innovation union in key areas. Hence, it will build a high-level innovation chain that serves the development of regionally characteristic advantage industries [1]. The growth enterprise market (GEM), also referred to as the secondary market, is an important supplement in the motherboard market. The GEM is a new market set up to meet the needs of entrepreneurship and innovation; it targets growth-oriented start-ups and focuses on supporting enterprises with independent innovation

abilities to go public. Since its opening in 2009, the GEM has actively responded to the national innovation-driven development strategy, playing a significant role in supporting the direct financing of enterprises and promoting innovation and development. It has also become an important gathering place for innovative and entrepreneurial enterprises in China. Therefore, optimal development of the GEM contributes significantly to the innovation and development of the YREB. The GEM has significant development potential. It is crucial for improving the innovation efficiency of companies listed on the GEM to drive the YREB to be the main force in high-quality economic development.

Under the background of globalization of economic activities, China's economy is in a stage of transition to high-quality development. Its connotation is to achieve high efficiency, and the essence of it is to promote the improvement of innovation efficiency. At present, international competition is increasingly fierce. Tracking and grasping the international innovation situation has also become the focus of research. From the perspective of the international innovation situation, national innovation efficiency shows the characteristics of spatial relevance and difference, and external environmental factors have a greater impact on national innovation efficiency. In recent years, China's overall innovation efficiency has been continuously improving [2], and the gap between China and developed countries is gradually narrowing. However, according to the current situation of China's innovation efficiency, there are still some problems, such as unbalanced regional development, large gaps in innovation efficiency between regions [3], and uneven levels of innovation efficiency among industries with different levels of resource allocation and utilization [4]. Listed companies, as the backbone of China's economy, are the core subjects of innovation-driven strategies, and technological innovation is the source of power for the survival and development of enterprises. In recent years, their innovation efficiency shows the following characteristics: overall technological innovation efficiency is low [5]; the heterogeneity effect of enterprise technological innovation efficiency is significant; ref. [6] and the heterogeneity of regions and industries is significant [7]. As a gathering place of high-tech and small and medium-sized enterprises, GEM-listed companies have great development potential and high innovation potential. The research on their innovation efficiency is more representative and of practical significance. As one of the important development areas in China, the Yangtze River Economic Zone will show what characteristics and development trends of its GEM-listed companies' innovation efficiency will be the focus of this paper. This paper also intends to focus on the above three characteristics and development trend of the Yangtze River Economic Zone GEM-listed companies to carry out research, for example: What is the Yangtze River Economic Zone GEM-listed companies' innovation efficiency status? What are the factors that affect its innovation efficiency? What are the characteristics of innovation efficiency among different industries and regions? What is the development trend of its innovation efficiency? Based on the above problems, this paper focuses on the research on the spatial and temporal evolution and development trend of innovation efficiency of the GEM-listed companies in the YREB.

Innovation efficiency has always been a research hotspot in academia. In recent years, domestic and international scholars focused on many dimensions of innovation efficiency. This study further critically evaluates some previous studies on the calculation and convergence trends of innovation efficiency. Scholars generally use two approaches to compute innovation efficiency: parametric and nonparametric methods. The parametric method mostly represents random front analysis (SFA), while the nonparametric method mostly represents data envelope analysis (DEA). Wadud and White (2000) [8] use the SFA and DEA methods, respectively, to calculate the efficiency of farmers in Bangladesh. The conclusions between the two methods are consistent. Leejy (2005) [9] further evaluated and compared the technical efficiency scores of 79 forest and paper firms using two methods. Bin (2015) [10], Yong (2015) [11], Fang (2020) [12], Ming et al. (2019) [13], and Iglesias, Castellanos, and Seijas (2010) [14] also calculated the technical efficiency and total factor productivity in different fields using the SFA method. Compared with the SFA method, the DEA method does not need to set a production function and is more advantageous in

managing the efficiency evaluation problem of multiple inputs and outputs [15]. Therefore, the DEA method is increasingly widely applied and continuously expanding. Li et al. (2010) [16] and other related studies constructed a resource allocation model based on the DEA method; Kádárová et al. (2015) [17] integrated the DEA and balanced scorecard methods to establish a comprehensive performance and efficiency management system for industrial enterprises and their processes; Genwen et al. (2017) [18] further conducted a comprehensive study on the three-stage DEA model; Na et al. (2019) [19], Tachega et al. (2021) [20], and other related studies used static DEA methods to calculate the innovation efficiency of China's environmental protection industry and the energy efficiency of African oil-producing economies, respectively; Wenjinget al. (2015) [21] and Shijian et al. (2018) [22] used the two-stage DEA method to compute innovation efficiency using interprovincial and industrial industry data, respectively; Pishgar-Komleh et al. (2020) [23] and other related studies further calculated the efficiency of the poor output of Polish winter wheat using the life cycle assessment + DEA framework; Huangbao (2014) [24] adopted the DEA Malmquist method to evaluate the innovation efficiency of Chinese high-technology industries. Regarding research on the convergence trend of innovation efficiency, most scholars conducted comprehensive studies on their convergence trends of innovation efficiency using macro inter-provincial and industry data from the regional [25–29] and industrial [30–33] levels, which provided a solid theoretical foundation for this study. However, there are few studies on the convergence of micro firm innovation efficiency in the existing literature. Therefore, this study further explores the convergence problem of innovation efficiency at the firm level on the basis of existing theoretical studies.

As one of the important development regions in China, the innovation of the YREB has received the attention and research of many scholars. Cheng, Y et al. [34] focused on the activities of entrepreneur financiers in the YREB, and made relevant research on financing innovation activities in the Yangtze River Delta region; Liu Y et al. [35] studied the spatial integration of urban agglomeration in the YREB and its relationship with industrial development, so as to provide support for future regional spatial integration and coordinated development; Zou L et al. [36] studied the efficiency of high-tech innovation in the region and found that the efficiency of high-tech innovation in China is continuously improving, but the regional differences are large, and put forward policy recommendations on the research conclusions; Meng D et al. [37] analyzed the spatial agglomeration effect of listed companies in China's YREB and its causes, and studied the main factors affecting their innovation efficiency. Yi M et al. [38] studied the impact of government research and development subsidies and environmental regulations on green innovation efficiency of the manufacturing industry in China's YREB; Li X et al. [39] took the Yangtze River Economic Zone as the research object to explore the impact of foreign direct investment on high-quality economic development under environmental regulations.

In summary, existing studies on innovation efficiency are relatively rich. However, there is still room for improvement and depth. The marginal contribution of this study, relative to what has been studied, may lie in the following considerations: (1) In the research sample, there is a plethora of literature investigating innovation efficiency in the macro region and industry. However, there is less literature on innovation efficiency at the micro firm level. Moreover, most previous studies focused on innovation efficiency in the entire YREB region or in some of these industries. In contrast, this study chose GEM-listed companies in the YREB as the research object. Data on micro enterprises are selected for empirical analysis, which provides a new research perspective for improving innovation efficiency in the YREB. (2) In terms of research content, the previous literature has little information on the industry heterogeneity and spatiotemporal evolution laws of corporate innovation efficiency; this study, based on previous research, further enriches the research content, joins the analysis of industry heterogeneity, and explores the laws of corporate innovation efficiency evolution over time, by conducting convergence tests to improve the research system regarding the innovation efficiency of enterprises, and ensures that it has significance for further improving the innovation efficiency of GEM-listed companies in

the YREB and other regions. This article proceeds as follows. Section two introduces the data sources and related model methods. Section three introduces the basic situation of GEM-listed companies in the YREB. Section four, which is the core of this paper, presents the empirical analysis. Section five presents the research conclusions and discussion of this article.

## 2. Materials and Methods

### *2.1. Data Sources*

Based on the latest cutting-edge research results in China and abroad, this study focuses on 142 GEM-listed companies in the YREB based on the availability and continuity of time for all types of data indicators. There are too much missing company patent data before 2015 and 2020, and only a few companies published them. Therefore, this study sets the data sample period as 2015–2019, with a sample size of 142. The sample data are mainly derived from the China Stock Market & Accounting Research (CSMAR) and Chinese Research Data Services Platform (CNRDS) databases, National Bureau of Statistics of China, and official website of the Shenzhen Stock Exchange (all companies in this paper are screened on this site).

### *2.2. Model Approach*

This paper first introduces the basic situation of the listed companies on the Growth Enterprise Market in the Yangtze River Economic Belt, then studies the present situation and development trend of their innovation efficiency, and finally puts forward a summary and relevant suggestions. This writing idea closely follows the research theme of the article and can highlight the research focus of the article. On the research method, using the kernel density analysis method and ArcGIS10.2 software to reflect the number changes of companies listed on the Growth Enterprise Market in the Yangtze River Economic Belt in recent years intuitively, and referring to the existing research [10,12,31], the DEA-BCC model and Malmquist index are used to analyze the current situation of innovation efficiency from static and dynamic dimensions. The $\sigma$ convergence and $\beta$ convergence analyses in this paper are used to track the change trend of innovation efficiency, while the Tobit model can be used to analyze the influencing factors of innovation efficiency. The specific models and methods are as follows.

#### 2.2.1. Dense Score Kernel Densitometry Method

This method uses a kernel function to calculate the amount per unit area from the point or broken-line elements, to fit individual points or broken lines to a smooth cone-like surface further. A kernel density analysis tool was used to calculate the density of the features in the surrounding neighborhood. This tool calculates both the density of point features and that of line features using the following equation:

$$f(s) = \sum_{i=1}^{n} \frac{1}{h^2} K \frac{s - c_i}{h} \tag{1}$$

where $f(s)$ is the kernel density calculation function at spatial position $s$; $h$ is the broad band; $n$ is the number of features that are less than or equal to $h$ from position $s$; and the $K$ function represents the spatial weight function. There are two key parameters for kernel density estimation: the spatial weight function $K$ and the distance decay threshold $h$ [40].

#### 2.2.2. DEA-BCC Model

There are two basic DEA models: the CCR and BCC models. The CCR model computes the relative efficiencies of inputs and outputs under conditions of constant returns to scale, while the BCC model computes the relative efficiencies of inputs and outputs under conditions of variable returns to scale. A firm's technological innovation is a complex systemic behavior with marginal benefit uncertainty [41]. Therefore, this study introduces

the output-oriented DEA-BCC model under variable scale conditions, to compute and evaluate the innovation efficiency during the period from 2015 through 2019 for GEM-listed firms in the YREB as follows:

$$
\begin{aligned}
min\theta &= \left[ \theta_0 - \varepsilon \left( \sum_{i=1}^{m} s_r^{+} + \sum_{i=1}^{m} s_r^{-} \right) \right] \\
s.t. \ & \sum_{j=1}^{m} \lambda_j X_{ij} + s_i^{-} = \theta_0 X_{i0} \\
& \sum_{j=1}^{m} \lambda_j Y_{ij} - s_i^{+} = Y_{r0} \\
& \sum_{j=1}^{n} \lambda_j = 1 \\
& \lambda_j \geq 0, s_r^{+} \geq 0, \ s_i^{-} \geq 0, \ i = 1, 2, \ldots, m \\
& r = 1, 2, \ldots, s \ j = 1, 2, \ldots, n,
\end{aligned}
\tag{2}
$$

where $\theta$ is the DEA technical efficiency score of the DMU (decision making unit), which takes values between zero and one. $s_i^{+}$ and $s_i^{-}$ indicate insufficient output and input redundancy, respectively. When $\theta$ is 1 and $s_i^{+} = s_i^{-} = 0$, the decision cell is a strong effective decision cell; when $s_i^{+}$ and $s_i^{-}$ are not equal to 0, it indicates that the decision-making unit is a weak effective unit, which may result in input redundancy and uneven resource allocation; when the value of $\theta$ is less than 1, it indicates that the decision-making unit is in an invalid state, and that the efficiency of resource allocation should be improved.

### 2.2.3. Malmquist Index

The Malmquist index, proposed by Malmquist, [42] is a dynamic analysis method developed based on the DEA model. Based on this theory and the DEA theory, Fare et al. expanded the application range of the Malmquist index in various fields [43]. The Malmquist index can be divided into technical efficiency (*EC*) and technical progress (*TC*) using the following relation:

$$
MI_{it} = EC_{it} \times TC_{it}
\tag{3}
$$

where $EC_{it}$ represents the movement of the production front face from period $t$ to $t + 1$, and $TC_{it}$ represents the catch-up speed of a certain DMU to the production likelihood boundary from period $t$ to $t + 1$. When the Malmquist index is larger than one, it indicates that the innovation efficiency of a firm as a whole shows an increasing trend over time. When the Malmquist index equals one, it indicates that the innovation efficiency of the firm as a whole does not change over time. When the Malmquist index is less than one, it indicates a decreasing trend in the innovation efficiency of firms as a whole. When a certain change ratio that constitutes the Malmquist index is greater than one, it indicates that this factor results in an increase in the level of innovation efficiency of firms.

### 2.2.4. $\sigma$-Convergence

To test whether the standard deviation of the innovation efficiency level of the GEM-listed firms in the YREB will decrease gradually over time, the $\sigma$-convergence coefficient was calculated to show a decreasing trend over time. The innovation efficiency level in this region is further scaled down over time compared with the overall innovation efficiency level. The calculation formula for $\sigma$-convergence is shown in Equation (4), where $\ln IE_{i,t}$ represents the natural log value of innovation efficiency in year $t$ for region $i$, and $N$ is the total number of regions.

$$
\sigma_t = \sqrt{\sum_i \frac{(\ln IE_{i,t} - \ln IE_t)^2}{N}}
\tag{4}
$$

### 2.2.5. $\beta$-Convergence

This study analyzes other changing laws of the evolution of the innovation efficiency of GEM-listed firms in the YREB over time, using the $\beta$ convergence test, which can be divided into $\beta$ absolute and conditional convergence, where $\beta$ absolute convergence refers to the growth rate of the innovation efficiency of GEM-listed firms in the YREB that is negatively related to their initial level, and, over time, innovation efficiency in all regions will reach the same steady-state level at some point in the future [44]. According to Barro and Sala-i-Martin's relevant research, the convergence model is shown in Equation (5).

$$\frac{1}{T}\ln(IE_{it}/IE_{i0}) = \alpha + \beta\ln(IE_{i0}) + \varepsilon_{it} \tag{5}$$

where $IE_{i0}$ denotes the value of innovation efficiency at the beginning of the period for GEM-listed firms in region $I$; $IE_{i0}$ denotes the innovation efficiency value in period t of GEM-listed firms in region $i$; $T$ is the time span of the observation period; $\alpha$ is a constant term; $\beta$ is the parameter value to be estimated. In other words, $\beta$ is the absolute convergence coefficient, and $\varepsilon$ is the random error term. In the case that $\beta$ is less than 0 and is significant, then there is $\beta$ absolute convergence in the innovation efficiency of GEM-listed firms in the Yangtze River economy and vice versa. However, in the actual research process, some scholars found that some regions have very different economic bases, levels of openness, and market structures, resulting in the final trend of innovation efficiency to focus on different equilibrium points [44]. To this end, Barro, Sala-i-Martin [45], and other scholars proposed the convergence and expansion of the $\beta$ condition, which considers the influence of other factors on the basis of the absolute convergence of $\beta$; a relevant control variable is added; and its model is shown in Equation (6).

$$\frac{1}{T}\ln(IE_{it}/IE_{i0}) = \alpha + \beta\ln(IE_{i0}) + \gamma X_{it} + \varepsilon_{it} \tag{6}$$

where $X_{it}$ denotes the other control variables added; $\beta$ is the parameter to be estimated; that is, $\beta$ is the conditional convergence coefficient; $\gamma$ denotes the coefficient of the control variables; and the means of the remaining variables and Equation (5) are similar. Through the signs and significance of $\beta$ and $\gamma$, we can ascertain whether each region converges to its own steady state and the main factors affecting its own convergence. Additionally, according to the estimated values of the $\beta$ absolute and conditional convergence coefficient coefficients, the convergence speed $S$ and half-life cycle $\tau$ of each region can be calculated; the specific calculation formulas are shown in Equations (7) and (8).

$$s = -\ln(1 - |\beta|)/T \tag{7}$$

$$\tau = \ln(2)/s \tag{8}$$

### 2.2.6. Tobit Regression Model

This study constructs a Tobit multiple regression model by taking the value of innovation efficiency of the GEM-listed companies of the YREB as the explanatory variable and the influencing factors of innovation efficiency of the companies listed on the GEM of each province and city as the explanatory variable. The calculation formula is:

$$Y = \begin{cases} Y_i = \beta_0 + \beta^t X_i + \mu_i, & Y_i > 0 \\ 0, & Y_i \leq 0 \end{cases} \tag{9}$$

## 3. Basic Information of GEM Listed Companies in the YREB

The data collection deadline was November 2021. Therefore, this study selected companies listed on the GEM of the YREB as the research object until November 2021. The relevant data of three representative years (2010, 2015, and 2020) in the early, middle,

and late stages of the sample are used to study the change in the number distribution of GEM-listed companies in the YREB.

### 3.1. Classified Statistics of GEM-Listed Companies in the YREB

This study divides the YREB into three regions: the Chengyu city cluster, the middle reaches of the Yangtze River, and the Yangtze River Delta. The number of companies listed on the GEM of the YREB until 2021 was counted. The Chengyu city cluster includes Yunnan, Guizhou, Sichuan, and Chongqing. The middle reaches of the Yangtze River include Hubei, Hunan, and Jiangxi provinces. The Yangtze River Delta includes Anhui, Zhejiang, Jiangsu, and Shanghai provinces.

By 2021, the Yangtze River Delta region had the largest number of GEM-listed companies (380), accounting for 75.40%. Jiangsu Province had the largest number of GEM-listed companies: 152,135 from Zhejiang Province, 66 from Shanghai, and 27 from Anhui Province. The number of GEM-listed companies in the provinces and cities in the middle reaches of the Yangtze River and Chengyu city cluster is relatively low, at 75 and 49, respectively, accounting for 14.88% and 9.72%, respectively. The number of GEM-listed companies in Yunnan, Chongqing, and Guizhou is relatively low, and the number of GEM-listed companies in Hubei, Hunan, Jiangxi, and Sichuan is similar and evenly distributed in Table 1.

**Table 1.** Number distribution of GEM-listed companies in the YREB.

| Chengyu City Cluster | Middle Reaches of the Yangtze River | Yangtze River Delta |
|:---:|:---:|:---:|
| Yunnan 5 | Hubei 28 | Anhui 27 |
| Guizhou 2 | Hunan 31 | Zhejiang 135 |
| Chongqing 5 | Jiangxi 16 | Jiangsu 152 |
| Sichuan 37 | | Shanghai 66 |
| Total 49 | Total 75 | Total 380 |
| | Sum 504 | |

### 3.2. The Spatiotemporal Evolution of GEM-Listed Companies in the YREB

This study examines the increase and decrease in the number and shift of the center of gravity of GEM-listed companies in the YREB since the opening of the GEM market in 2009 to reflect the spatiotemporal evolution characteristics of the number of GEM-listed companies in 11 provinces and cities in the YREB directly. The relevant data for the early, middle, and late years (2009, 2015, and 2021, respectively) were selected. Based on the administrative map of the YREB, ArcGIS10.2 software was used to conduct visual processing and kernel density analysis on the number distribution of GEM-listed companies in 11 provinces and cities of the YREB, as shown in Figure 1. Overall, the number of GEM-listed companies in the YREB increased significantly during the three periods. Among the four provinces and cities in the Chengyu city cluster, the number of companies in Sichuan and Chongqing developed the fastest. However, the number of companies in Guizhou and Yunnan developed slowly. Among the three provinces in the middle reaches of the Yangtze River, Hubei and Hunan developed faster and distributed more companies than Jiangxi. The number of companies in the Yangtze River Delta increased the most obviously among the three regions; they were mainly concentrated in Jiangsu, Shanghai, and Zhejiang provinces. In contrast, there were relatively few companies in Anhui province. The reason for the significant increase in the number of enterprises is that, on one hand, the Chinese government issued many preferential and supportive policies for domestic small and medium-sized enterprises, which boosted the rapid growth of a large number of enterprises. On the other hand, the development of science and technology made more and more people choose to start their own businesses, and the way in which enterprises are set up is more flexible.

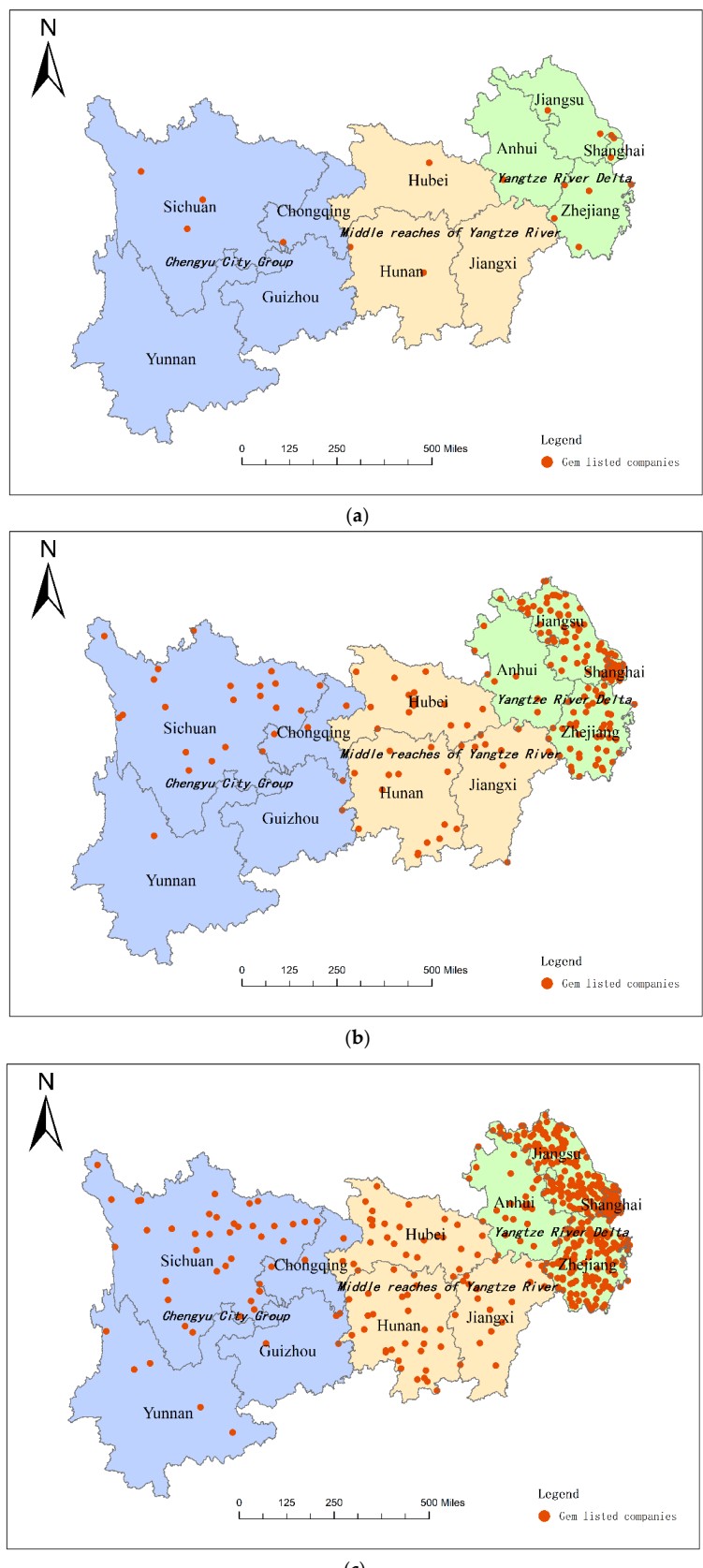

**Figure 1.** Spatial-temporal evolution of GEM-listed companies in the YREB. (**a**) 2009; (**b**) 2015; (**c**) 2021.

## 4. Results

### 4.1. Data Processing

The original samples were screened scientifically in this study. First, ST (shares in which the company suffered losses for two consecutive years and which require special treatment) and *ST companies (shares of the company that suffered losses for three consecutive years and were given early warning of delisting) are excluded. These companies have financial anomalies, which may affect the reliability of their evaluation results. Second, to ensure the accuracy of the empirical results, this study deleted companies with a large amount of missing input-output index data according to the availability of various data indicators and the continuity of time. In conclusion, 142 GEM-listed companies in the YREB are identified, including 21 in the Chengyu city cluster, 27 in the middle reaches of the Yangtze River, and 94 in the Yangtze River Delta. This study divides the YREB into three regions to explore the spatiotemporal transition and convergence of innovation efficiency. To improve the reliability of the data analysis results, the following two pretreatments were performed on the collected sample data before conducting the empirical analysis.

- Irrational missing data processing. To ensure data integrity, this study first deleted company samples with a large amount of missing input-output index data. For patent application data, this study used the corresponding data in the CNRDS database to match. The rest of the input-output index data were obtained from the CSMAR database.

- Dimensionless processing. On the one hand, because the DEA model can only identify non-negative data in the calculation process, there are a few negative numbers in the original net profit and operating income data. On the other hand, there is a significant difference between the values of different indicators in the original data of this study. If the calculation is performed directly, the effect of small values can be ignored, resulting in inaccurate calculation results. Considering these two factors, we normalized the original data; the processing formula is as follows:

$$X^* = 0.1 + 0.9 \times \frac{X_i - \min(X_i)}{\max(X_i) - \min(X_i)} \tag{10}$$

In this equation, $X^*$ and $X_i$ are the standardized and original data, respectively. After dimensionless processing, all the values are evenly distributed between 0.1 and 1; values only shift or scale, and the shape of the production front surface does not change. Therefore, it not only ensures that the data conform to the operation rules, but also does not affect the empirical results.

### 4.2. Index Selection

Based on the research results of the current academic circle on the innovation efficiency evaluation index system [46,47], the number of R&D personnel and the amount of R&D investment are selected as the input indicators of the innovation efficiency evaluation system of the GEM-listed companies in the YREB after comprehensively considering the coverage of index content and data availability. Net profit, operating income, and the number of patent applications are output indicators. Table 2 shows the specific evaluation index systems.

**Table 2.** Evaluation index of the innovation efficiency of GEM-listed companies in the YREB.

| Indicators | The Variable Name | Indicator Description |
|:---:|:---:|:---:|
| Input indicators | A1 | Number of R&D personnel (persons) |
| | A2 | R&D Investment (YUAN) |
| Output indicators | B1 | Net profit (YUAN) |
| | B2 | Operating income (YUAN) |
| | B3 | Number of patent Applications (pieces) |

In the process of analyzing the influencing factors of innovation efficiency, through combing the relevant literature in recent years and considering the representativeness of the selected indicators and the availability of data, this study selects the innovation efficiency of 142 GEM-listed companies in 11 provinces and cities in the YREB from 2015 to 2019 as the explanatory variable ($Y$), and selects regional GDP ($X_1$), R&D funding intensity ($X_2$), foreign direct investment ($X_3$), and the number of listed companies ($X_4$) as the explanatory variables from the perspectives of economic foundation, government support, openness to the outside world, and market structure.

### 4.3. Spatiotemporal Transition of the Innovation Efficiency of GEM-Listed Companies in the YREB

#### 4.3.1. Static Analysis of Innovation Efficiency

This study uses Deap2.1 software to analyze the innovation efficiency of the input and output indices of GEM-listed companies in the YREB from 2015 to 2019 comprehensively. Appendix A shows the innovation efficiency values of the 142 GEM-listed companies. Meanwhile, this study divides the YREB into three regions and calculates the change in innovation efficiency and the mean value of comprehensive technological innovation efficiency of GEM-listed companies in the three regions from 2015 to 2019. Table 3 shows the relevant results. On this basis, this study uses the form of image color gradient to reflect the mean difference in the innovation efficiency of GEM-listed companies in different regions during the mentioned sample period intuitively. Figure 2 shows the results. Overall, the mean values of innovation efficiency among the three regions of the YREB differ significantly and show the spatial differentiation characteristics of provinces in the middle reaches of the Yangtze River > Chengyu city cluster > the Yangtze River Delta region. The mean value of innovation efficiency is easily affected by extreme values owing to the large number of GEM-listed companies in the Yangtze River Delta region. This further results in an overall low mean value of innovation efficiency. In contrast, the innovation efficiency of GEM-listed companies in the middle reaches of the Yangtze River was stable at about 0.8 and maintained a slight increase in 2015 and 2019. However, the overall efficiency value did not exceed 1, indicating that the innovation and development level of GEM-listed companies in all regions still has room for improvement.

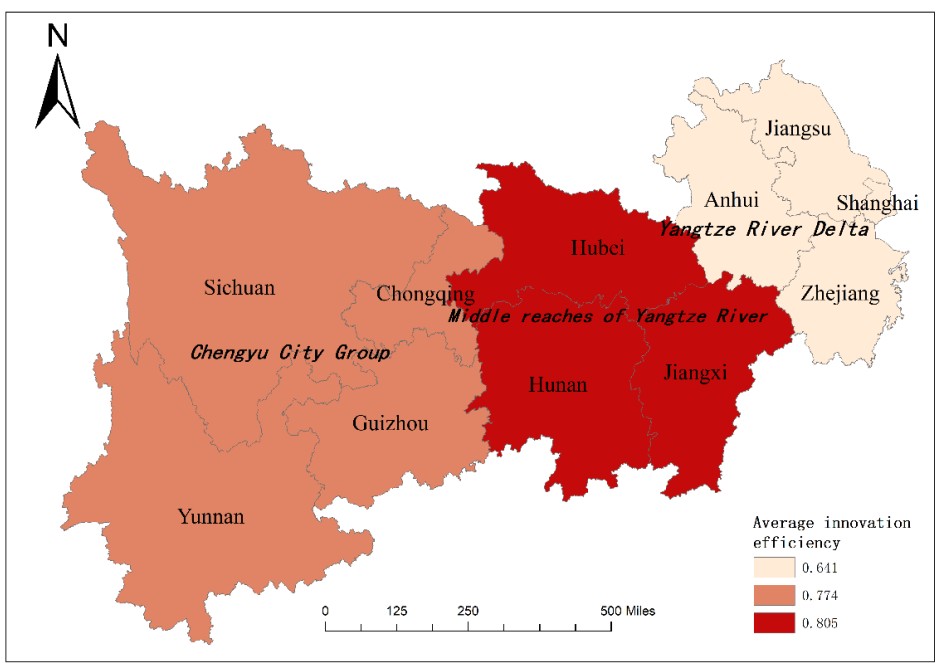

**Figure 2.** Average innovation efficiency of GEM-listed companies in the middle reaches of the Yangtze River from 2015 to 2019.

**Table 3.** Average innovation efficiency of the GEM-listed companies in the Yangtze Economic Belt by region.

| Regions | 2015 | 2016 | 2017 | 2018 | 2019 | Mean |
|---|---|---|---|---|---|---|
| Chengyu city cluster | 0.817 | 0.793 | 0.721 | 0.804 | 0.734 | 0.774 |
| the middle reaches of the Yangtze River | 0.837 | 0.857 | 0.746 | 0.775 | 0.810 | 0.805 |
| the Yangtze River Delta | 0.626 | 0.629 | 0.577 | 0.678 | 0.695 | 0.641 |
| Mean | 0.760 | 0.759 | 0.681 | 0.752 | 0.746 | 0.740 |

4.3.2. Dynamic Analysis of Innovation Efficiency

Based on the previous scholars' research [48,49], this study uses Deap2.1 software and the DEA-Malmquist model to evaluate the dynamic efficiency changes of 142 GEM-listed companies from 2015 to 2019 in the Chengyu city cluster, provinces, and cities in the middle reaches of the Yangtze River and the Yangtze River Delta region. This study explores the innovation efficiency of GEM-listed companies in the YREB from 2015 to 2019 from a dynamic perspective, analyzes the development trend of innovation efficiency during the evaluation period, and evaluates the determinants of total factor productivity growth. Appendix B shows the data of the MI Index and the decomposition quantity of the innovation efficiency of 142 GEM-listed companies. The annual average results of the Malmquist index in the different regions are as follows Tables 4–6.

**Table 4.** Annual average of the Malmquist Index of the GEM-listed companies in the Chengyu city cluster.

| Years | Technical Efficiency | Advances in Technology | Pure Technical Efficiency | The Scale Efficiency | Malmquist Index |
|---|---|---|---|---|---|
| 2015–2016 | 0.955 | 0.706 | 0.886 | 1.078 | 0.674 |
| 2016–2017 | 0.894 | 1.575 | 1.100 | 0.812 | 1.407 |
| 2017–2018 | 1.146 | 0.548 | 1.013 | 1.131 | 0.628 |
| 2018–2019 | 0.890 | 0.960 | 0.959 | 0.929 | 0.855 |
| Mean | 0.966 | 0.874 | 0.986 | 0.979 | 0.844 |

**Table 5.** Annual average of the Malmquist Index of GEM-listed companies in the provinces and cities in the middle reaches of Yangtze River.

| Years | Technical Efficiency | Advances in Technology | Pure Technical Efficiency | The Scale Efficiency | Malmquist Index |
|---|---|---|---|---|---|
| 2015–2016 | 1.028 | 0.971 | 1.008 | 1.020 | 0.999 |
| 2016–2017 | 0.852 | 1.136 | 0.859 | 0.991 | 0.967 |
| 2017–2018 | 1.039 | 1.773 | 1.037 | 1.002 | 1.841 |
| 2018–2019 | 1.050 | 0.717 | 1.115 | 0.941 | 0.753 |
| Mean | 0.989 | 1.088 | 1.000 | 0.988 | 1.076 |

**Table 6.** Annual average of the Malmquist Index of GEM-listed companies in Yangtze River Delta region.

| Years | Technical Efficiency | Advances in Technology | Pure Technical Efficiency | The Scale Efficiency | Malmquist Index |
|---|---|---|---|---|---|
| 2015–2016 | 1.017 | 1.183 | 1.020 | 0.997 | 1.203 |
| 2016–2017 | 0.897 | 1.201 | 0.874 | 1.027 | 1.077 |
| 2017–2018 | 1.194 | 1.251 | 1.336 | 0.894 | 1.493 |
| 2018–2019 | 1.046 | 1.048 | 1.046 | 1.000 | 1.097 |
| Mean | 1.033 | 1.168 | 1.056 | 0.978 | 1.207 |

The mean value of the Malmquist index and its decomposition value for the three regions show the spatial differentiation characteristics of the Yangtze River Delta region > provinces in the middle reaches of the Yangtze River > Chengyu city cluster. The overall innovation efficiency of GEM-listed companies in the middle reaches of the Yangtze River Delta is also improving steadily and slowly, with a growth rate of 7.6%. However, the Malmquist index of the Chengyu city cluster is 0.844, which indicates that the innovation efficiency of the companies listed on the GEM in the Chengdu-Chongqing region rather decreased from 2015 to 2019, and the reasons need to be considered.

Analysis of the expansion of the Malmquist index: Regarding the Yangtze River Delta region, the technical efficiency is 1.033, indicating that effective management methods and decision-making means improved technical efficiency. The pure technical efficiency (PTE) is 1.056, indicating that the application level of technology is constantly improving, while the scale efficiency is 0.978, indicating that the scale has not reached the optimal level, and technological progress is 1.168. This shows that technology significantly improved and progressed there. The improvement in technical efficiency emanates from improvement in the PTE. Technical efficiency and technological progress jointly promote improvement in the overall efficiency. In the middle reaches of the Yangtze River, technical efficiency decreases as technological progress increases. The overall efficiency improvement is a result of an increase in technological progress. However, scale efficiency needs to be improved. Regarding the Chengyu city cluster, the overall innovation efficiency is low, and the decomposition value of each part is less than 1, indicating that technical efficiency, scale efficiency, and technological progress are not optimal, and further improvement is needed.

According to the line chart of the annual change in the MI index (Figure 3), the MI index of the three regions of the YREB fluctuates significantly, while rising first and then falling. During the period of 2017–2018, the MI index of the middle reaches of the Yangtze River and the delta region reached its peak, while the MI index of the Chengyu city cluster reached its lowest. The reason for this difference can be seen from the decomposition value of the MI index, which is mainly due to the large difference in the decomposition value of technological progress, which represents the same input but results in more outputs due to technological progress. In fact, this index represents the degree of technological innovation. From 2017 to 2018, the middle reaches of the Yangtze River continued to make efforts and innovations, which resulted in continuous technological innovation and upgrading. The index of technological progress reached 1.773, and under the original technological conditions, the input-output efficiency was also improved. During this period, the efficiency of the Yangtze River Delta region improved to varying degrees under both the original technological level and the new technological conditions, but the improvement rate is not as high as that of the middle reaches of the Yangtze River. The Chengyu city cluster is obviously inferior in exploring technological innovation, with a technological progress index value of only 0.548. Although the region's efficiency also improved under the original technological level, the region's new technology is not mature enough to bring its efficiency into full play, resulting in a significant difference in innovation efficiency between the region and the other two regions. Although the overall MI index of the Chengyu city cluster was low, it showed a rising trend since 2017. The Yangtze River Delta and the middle reaches of the Yangtze River experience a decline in the MI index during this period, which provides a good opportunity for GEM-listed companies in the Chengyu city cluster to catch up with other regions. From 2018 to 2019, the three regions tended to be consistent and gradually approached 1.

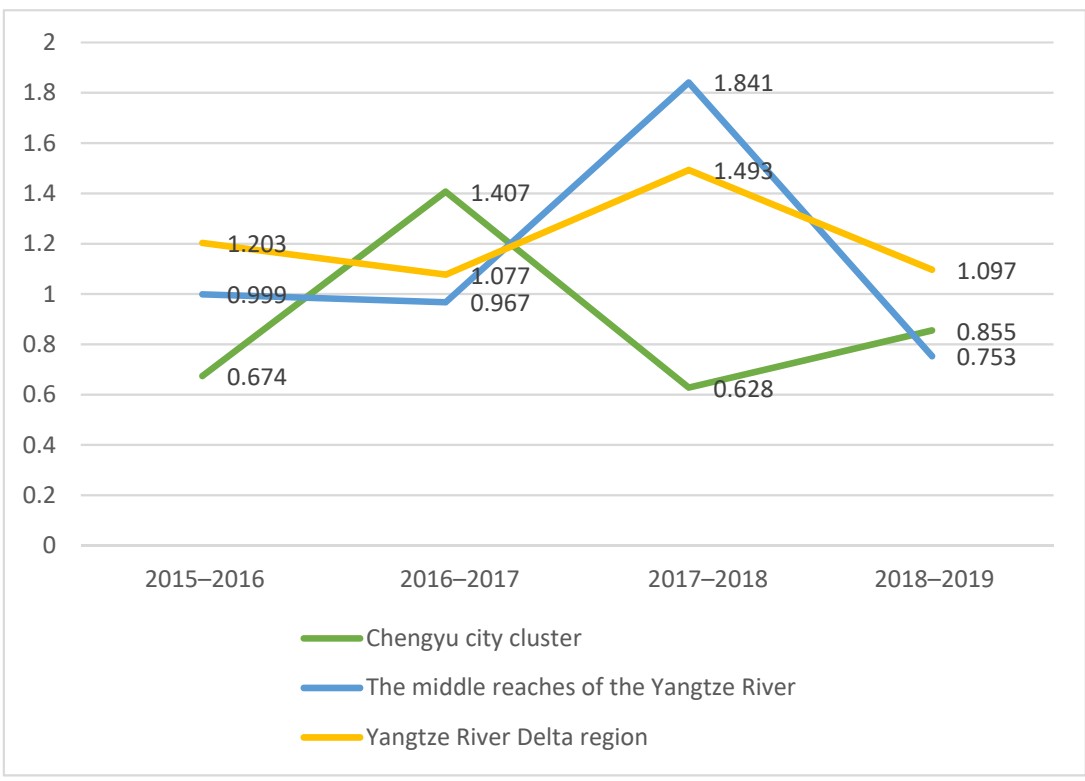

**Figure 3.** Variation trend of annual mean of Malmquist index by region.

### 4.4. Industry Heterogeneity Analysis of Innovation Efficiency

GEM-listed companies in the industry differ significantly. The company's business objectives and business philosophy also differ. According to statistics, most GEM-listed companies in the YREB are concentrated in industry and public utilities, whereas a few are concentrated in education, commerce, real estate, and other industries, which can be further subdivided into different industries. Analyzing the concentration of GEM-listed companies in different industries and the change in their innovation efficiency is conducive to proposing targeted countermeasures and suggestions for different industries. Table 7 shows the specific results.

According to Table 7, the GEM-listed companies of the YREB are distributed in 29 industries, with obvious industry agglomeration effects. Among the 142 enterprises, 21 were special equipment manufacturing; 20 were computer, communication, and other electronic equipment manufacturing; 19 were electrical machinery and equipment manufacturing; 12 were software and information technology services; and 11 were pharmaceutical manufacturing. In other words, most GEM-listed companies belong to the industrial and public utility industries, while other industries are fewer and scattered. Regarding innovation efficiency, industries with lower competitive strength have higher innovation efficiency, among which that of the wholesale industry is as high as 0.994; that of the radio, television, film, and television recording industries are as high as 0.929; that of the business service industry is as high as 0.972; that of the oil and natural gas mining industry is as high as 0.945; that of the metal products industry is as high as 0.969. These industries have lower competition levels and industry barriers. Companies that enter the GEM market may consider entering these industries.

**Table 7.** Industry heterogeneity analysis of the innovation efficiency of GEM-listed companies in the YREB.

| Industry Name | Number of Industries | 2015 | 2016 | 2017 | 2018 | 2019 | Mean |
|---|---|---|---|---|---|---|---|
| Special equipment manufacturing | 21 | 0.752 | 0.738 | 0.707 | 0.785 | 0.772 | 0.751 |
| Nonferrous metal smelting and rolling processing industry | 1 | 0.479 | 0.765 | 0.882 | 0.837 | 0.791 | 0.751 |
| Pharmaceutical manufacturing | 11 | 0.786 | 0.781 | 0.613 | 0.827 | 0.715 | 0.744 |
| General equipment manufacturing | 9 | 0.748 | 0.738 | 0.652 | 0.648 | 0.686 | 0.695 |
| Automobile industry | 3 | 0.767 | 0.750 | 0.578 | 0.682 | 0.717 | 0.699 |
| Manufacturing of computers, communications, and other electronic equipment | 20 | 0.703 | 0.694 | 0.667 | 0.665 | 0.694 | 0.685 |
| Chemical raw materials and chemical products manufacturing | 6 | 0.769 | 0.768 | 0.666 | 0.752 | 0.777 | 0.746 |
| Instrument manufacturing | 6 | 0.612 | 0.634 | 0.462 | 0.559 | 0.630 | 0.579 |
| Manufacturing of railway, ship, aerospace, and other transportation equipment | 1 | 0.760 | 0.729 | 0.580 | 0.825 | 0.770 | 0.733 |
| Metal products industry | 1 | 0.906 | 0.940 | 1.000 | 1.000 | 1.000 | 0.969 |
| Nonmetallic mineral products industry | 4 | 0.654 | 0.672 | 0.574 | 0.757 | 0.803 | 0.692 |
| Electrical machinery and equipment manufacturing | 19 | 0.619 | 0.623 | 0.588 | 0.672 | 0.713 | 0.643 |
| Rubber and plastic products industry | 5 | 0.595 | 0.6242 | 0.654 | 0.771 | 0.773 | 0.683 |
| Gas production and supply industry | 1 | 0.848 | 0.899 | 0.915 | 0.858 | 0.871 | 0.878 |
| Oil and gas extraction industry | 1 | 0.987 | 1.000 | 0.959 | 0.975 | 0.802 | 0.945 |
| Agriculture | 1 | 0.582 | 0.632 | 0.482 | 0.559 | 0.690 | 0.589 |
| Software and information technology services | 12 | 0.551 | 0.530 | 0.449 | 0.535 | 0.557 | 0.524 |
| Telecommunications, radio and television, and satellite transmission services | 1 | 0.624 | 0.672 | 0.842 | 1.000 | 0.930 | 0.814 |
| Ecological protection and environmental governance | 5 | 0.730 | 0.676 | 0.616 | 0.699 | 0.823 | 0.709 |
| Professional and technical services | 3 | 0.600 | 0.585 | 0.493 | 0.597 | 0.631 | 0.581 |
| Hygiene | 1 | 0.406 | 0.566 | 0.48 | 0.654 | 0.920 | 0.605 |
| Business services | 1 | 0.869 | 1.000 | 0.99 | 1.000 | 1.000 | 0.972 |
| Internet and related services | 2 | 0.517 | 0.600 | 0.472 | 0.636 | 0.624 | 0.569 |
| Radio, television, film, and television recording production industry | 1 | 0.800 | 0.845 | 1.000 | 1.000 | 1.000 | 0.929 |
| Storage industry | 1 | 0.660 | 0.623 | 0.499 | 0.584 | 0.585 | 0.590 |
| Wholesale industry | 2 | 1.000 | 1.000 | 1.000 | 1.000 | 0.970 | 0.994 |
| Retail | 1 | 0.572 | 0.650 | 0.501 | 0.542 | 0.860 | 0.625 |
| Public utility | 1 | 0.833 | 1.000 | 0.851 | 0.889 | 0.385 | 0.792 |
| Civil engineering and construction | 1 | 1.000 | 1.000 | 0.702 | 0.581 | 0.714 | 0.799 |

*4.5. Convergence Test Results*

Calculating the innovation efficiency of GEM-listed companies in 11 provinces and cities of the YREB shows that there are significant differences in the innovation efficiency of different provinces and cities. How does this difference change over time? Based on a dynamic analysis of the innovation efficiency of the Yangtze River economy, this study conducts a convergence test to analyze the evolution trend of the innovation efficiency of GEM-listed companies in the YREB further.

According to the formula of $\sigma$-convergence, the $\sigma$-convergence index values of innovation efficiency in the YREB, Chengyu city cluster, middle reaches of the Yangtze River region, and delta region are calculated. Table 8 shows the calculation results. From 2015 to 2019, the $\sigma$-convergence index value of the overall innovation efficiency of the YREB

first decreased to 0.2951 from 0.3064 and further increased to 0.3509. Subsequently, the $\sigma$-convergence index value has been on a downward trend for two years, generally showing an inverted N shape. Moreover, there is no $\sigma$-convergence trend. In other words, the difference in the overall innovation efficiency level of GEM-listed companies of the YREB did not decrease over time. The $\sigma$-convergence index value of innovation efficiency in the Chengyu city cluster during the period from 2015 to 2019 was generally N-shaped. However, there was no $\sigma$-convergence trend. The variation law of the $\sigma$-convergence index values in the Yangtze River Delta and the middle reaches of the Yangtze River from 2015 to 2019 is roughly similar to that of the YREB as a whole, and does not conform to the $\sigma$ convergence trend. This shows that there is no $\sigma$-convergence trend in the YREB as a whole or in its subregions. Although the $\sigma$-convergence index decreased in some years, it soon showed an upward trend; in other words, the $\sigma$-convergence index fluctuated continuously. Moreover, there was no obvious convergence trend. This shows that the deviation in the innovation efficiency level of listed companies in the GEM of the YREB as a whole and its subregions does not show a decreasing trend as time goes by.

**Table 8.** Innovation efficiency of GEM-listed companies of the YREB $\sigma$-convergence index value.

| Years | Yangtze River Economic Belt | Chengyu City Cluster | Yangtze River delta | Middle Reaches of the Yangtze River |
|---|---|---|---|---|
| 2015 | 0.3064 | 0.2300 | 0.3181 | 0.1926 |
| 2016 | 0.2951 | 0.2749 | 0.2899 | 0.1774 |
| 2017 | 0.3509 | 0.3257 | 0.3562 | 0.2727 |
| 2018 | 0.2925 | 0.2563 | 0.2994 | 0.2647 |
| 2019 | 0.2502 | 0.3263 | 0.2233 | 0.2348 |

Equation (4) was used to test the $\beta$ absolute convergence of the YREB; Table 9 shows the test results. The table clearly shows that the $\beta$ absolute convergence coefficient is less than 0 in both the YREB region and Chengyu city cluster, the middle reaches of the Yangtze River, and the Yangtze River Delta region, and is significant. There is a $\beta$ absolute convergence trend in these regions; in other words, the innovation efficiency of regions with low innovation efficiency will grow faster than those with high innovation efficiency over time and eventually converge to the same steady-state level at a certain point in time. Simultaneously, the convergence rates and semi-life cycles of different regions were calculated using Equations (7) and (8). The calculation results in the table show that the convergence rates and semi-life cycles of the YREB as a whole, Chengyu city cluster, Yangtze River Delta, and the middle reaches of the Yangtze River are 0.1139, 0.0491, 0.1281, and 0.1892, respectively, and 6.0835, 14.1087, 5.4089, and 3.6637 years, respectively. This clearly shows that the convergence rate of the Chengyu city cluster is slow, and the semi-life cycle is relatively long, whereas the convergence rate of the middle reaches of the Yangtze River is relatively fast, and the semi-life cycle is relatively short.

**Table 9.** Innovation efficiency of GEM-listed companies of the YREB $\beta$ Absolute convergence test.

| Regression Coefficient | Yangtze River Economic Belt | Chengdu Chongqing Region | Yangtze River Delta | Middle Reaches of the Yangtze River |
|---|---|---|---|---|
| $\alpha$ | −0.1772 *** (0.000) | −0.0978 *** (0.001) | −0.2042 *** (0.000) | −0.1857 *** (0.000) |
| $\beta$ | −0.4343 *** (0.000) | −0.2178 ** (0.049) | −0.4731 *** (0.000) | −0.6117 *** (0.000) |
| $s$ | 0.1139 | 0.0491 | 0.1281 | 0.1892 |
| $\tau$ | 6.0835 | 14.1087 | 5.4089 | 3.6637 |
| $R^2$ | 0.2491 | 0.0787 | 0.2732 | 0.2686 |
| $F$ | 69.83 | 4.00 | 47.50 | 24.91 |

Note: **, and *** are significant at 5%, and 1% levels, respectively. The values in brackets are $p$-values.

The $\beta$ absolute convergence test shows that there is a significant $\beta$ absolute convergence trend in the YREB as a whole, the middle reaches of the Yangtze River, and the delta region. The existence of the $\beta$-conditional convergence trend in these regions is further investigated by adding control variables to the regions with an absolute $\beta$-convergence trend. In this study, referring to the existing research literature [50,51], foreign direct investment (*fdi*), government support (*gov*), market competition degree (*mar*), infrastructure (*inf*), and innovation culture (*inc*) are selected as control variables, in which the proportion of total foreign direct investment to regional GDP represents foreign direct investment. The proportion of R&D funds to general government budget expenditure is regarded as government support. The degree of market competition, infrastructure, and innovation culture are expressed by the number of enterprises in the market, the proportion of total post and telecommunications business to GDP, and the proportion of venture capital to GDP, respectively. Formula (6) is used to calculate the conditional $\beta$-convergence coefficient and ascertain whether there is a conditional $\beta$-convergence trend in the entire YREB and each region. Table 10 shows the calculation results. The table clearly shows that the conditional $\beta$ convergence coefficients of the YREB as a whole, Chengyu city cluster, the middle reaches of the Yangtze River, and the delta regions are all less than 0, and are significant at 5%, which indicates that there is a trend of conditional $\beta$-convergence in these regions. Compared with the absolute $\beta$-convergence model, the goodness of fit of the model is improved after the control variables are added to the whole YREB, Chengyu city cluster, the middle reaches of the Yangtze River, and the delta regions; in other words, the explanatory ability of the conditional $\beta$-convergence model is enhanced.

**Table 10.** Convergence test of the $\beta$ condition of the innovation efficiency of GEM-listed companies in the YREB.

| Variable | Yangtze River Economic Belt | Chengyu City Cluster | Yangtze River Delta | Middle Reaches of the Yangtze River |
|---|---|---|---|---|
| $\alpha$ | −1.5531 *** | −1.8812 *** | −9.9408 *** | 13.4329 |
| | (0.004) | (0.008) | (0.000) | (0.557) |
| $\beta$ | −0.3871 *** | −0.2123 ** | −0.4281 *** | −0.4860 *** |
| | (0.000) | (0.028) | (0.000) | (0.001) |
| $fdi$ | 0.1708 *** | 0.0876 | 0.8190 *** | −0.7925 |
| | (0.001) | (0.191) | (0.000) | (0.613) |
| $gov$ | −0.1354 | −0.0460 | −0.6597 ** | 4.2828 * |
| | (0.106) | (0.560) | (0.047) | (0.063) |
| $mar$ | 0.0137 | −0.1758 ** | 0.3749 | −1.3378 |
| | (0.891) | (0.011) | (0.356) | (0.379) |
| $inf$ | 0.1903 *** | 0.0028 | 0.2248 *** | 0.1866 |
| | (0.000) | (0.960) | (0.000) | (0.631) |
| $inc$ | −0.0196 | 0.1074 * | −0.1309 | −0.2808 |
| | (0.731) | (0.058) | (0.426) | (0.229) |
| $s$ | 0.0979 | 0.0477 | 0.1118 | 0.1331 |
| $\tau$ | 7.0794 | 14.5230 | 6.2022 | 5.2075 |
| $R^2$ | 0.3638 | 0.2051 | 0.4507 | 0.3985 |
| $F$ | 24.19 | 3.17 | 21.81 | 12.62 |

Note: *, **, and *** are significant at 10%, 5%, and 1% levels, respectively. The values in brackets are *p*-values.

From the perspective of control variables, at the overall level of the YREB, both foreign direct investment and infrastructure investment are significant, with coefficients of 0.1708 and 0.1903, respectively, indicating that foreign direct investment and infrastructure investment have significant positive effects on the growth of innovation efficiency in GEM-listed companies in the YREB. Regarding infrastructure, the more optimal the regional infrastructure, the better the external conditions and operating environment that can be provided for enterprises, thus further promoting their innovation efficiency. Regarding foreign direct investment, on the one hand, it can increase market vitality, bring advanced technical knowledge and management experience, promote the imitation and learning of local enterprises, and improve their own innovation efficiency; on the other hand, the entry

of foreign enterprises will also result in competitive pressure on local enterprises. Local enterprises will further reinforce their R&D and innovation capabilities to enhance their competitiveness in the market, which will also force enterprises to improve their innovation efficiency. At the sub-regional level of the YREB, the degree of market competition in the Chengyu city cluster, that is, the number of enterprises in the market, will have a significant negative impact on innovation efficiency. The economic base and infrastructure in the Chengyu city cluster are relatively suboptimal. Therefore, the more enterprises in the market, the fewer resources a single enterprise can occupy, which is not conducive to the improvement of enterprise innovation efficiency, while the innovation culture will have a significant positive impact on innovation efficiency in this region. Compared with companies in other regions of the YREB, companies in the Chengyu City cluster have relatively immature external market conditions and need to increase innovation investment within enterprises to improve innovation efficiency through continuous exploration. Government support in the middle reaches of the Yangtze River had a significant positive impact on the region. Enterprises in the middle reaches of the Yangtze River are experiencing rapid development. The greater the government support for enterprise R&D, the better the innovation efficiency of enterprises. Foreign direct investment and infrastructure in the Yangtze River Delta will have a significant positive impact on innovation efficiency, while government support will have a significant negative impact on innovation efficiency. It may be due to the relatively rapid development of companies in the Yangtze River Delta region due to the advantages of location and resources, resulting in information asymmetry between the government and the subsidized enterprises, etc. Therefore, the government's support for enterprise innovation activities is inefficient. This clearly demonstrates the differences in geographical location, economic development, and factor endowments of different regions and indicates that the influence of external factors on the improvement of innovation efficiency in different regions differs.

### 4.6. Analysis of Factors Influencing Innovation Efficiency

From the regression results of Tobit model in Table 11, it can be seen that the overall economic scale and government research and development expenditure have a positive effect on the innovation efficiency of GEM-listed companies. The size of the listed companies has a negative effect on the innovation efficiency of the GEM-listed companies to a certain extent. Foreign direct investment has no significant effect on the innovation efficiency of the GEM-listed companies.

**Table 11.** Tobit model regression results.

| Y | Indicator Meaning | Coef. | Std. Err. | t | p > \|t\| |
|---|---|---|---|---|---|
| $X_1$ | Regional GDP | 0.000387 | $8.08 \times 10^{-5}$ | 4.79 | 0 |
| $X_2$ | R&D funding intensity | 0.02524 | 0.005416 | −4.66 | 0 |
| $X_3$ | Foreign direct investment | 0.000423 | 0.000314 | 1.35 | 0.178 |
| $X_4$ | The number of listed companies | −0.00044 | 0.00013 | −3.41 | 0.001 |
| _cons | | 0.808451 | 0.025519 | 31.68 | 0 |

Among them, the *p* value of the overall economic scale and the government's R&D expenditure is equal to 0, which indicates that the government has played a leading role in improving the innovation efficiency of the GEM-listed companies. The government's support and investment have aroused the attention of the small- and medium-sized enterprises to the research and development innovation, thus improving the innovation ability of the small- and medium-sized enterprises. At the same time, it indicates that the scientific and technological innovation funds of the enterprises are to a large extent derived from the government's investment, which also indicates that the marketization degree of the

scientific and technological innovation of the GEM-listed companies Regional GDP has a significant positive impact on the innovation efficiency of GEM-listed companies, which indicates that the substantial improvement of regional economic strength provides innovative capital sources for small and medium-sized GEM-listed companies, thus improving the innovation efficiency of GEM-listed companies, but the economic development has not produced obvious economies of scale. The scale of GEM-listed companies has a significant negative impact on their innovation efficiency, which may be because, when the GEM market is small, the innovation efficiency value is not easily affected by extreme values. With the increase of GEM market scale and innovation investment, there will inevitably be a low ratio of innovation output to investment, that is, ineffective utilization. GEM-listed companies should adjust their scale according to the market environment to improve the input utilization rate. Foreign direct investment failed the hypothesis test.

## 5. Conclusions and Discussion

### 5.1. Discussion

This study empirically examines the innovation efficiency of GEM-listed companies in the YREB. Innovation and improvement were made in research samples and research methods, revealing the current situation, characteristics, and trends of the development of GEM-listed companies in the YREB, and the assumptions mentioned above were confirmed. The premise of the sustainable development of the YREB is the balanced development between the region and the industries. In the process of empirical research, this study found that there is a large gap in innovation efficiency between the eastern and western regions of the GEM-listed companies of the YREB, and there is heterogeneity in industry innovation efficiency. These findings provide an improved thinking and direction for the sustainable development of the YREB in the future. At the same time, as an important economic entity of the YREB, the improvement in innovation efficiency of the GEM-listed companies can continuously promote the development of the YREB towards high quality. The static results show that the average innovation efficiency is 0.74, which further shows that there is still room for improvement. From a dynamic point of view, the scale efficiency of the three regions is not optimal. This indicates that the GEM-listed companies in the YREB should focus on the improvement of enterprise scale efficiency. The overall efficiency of GEM-listed companies in the delta region and the middle reaches of the Yangtze River is increasing, mainly because of technological progress; in other words, if enterprises can further realize technological innovation, then, improvement in innovation efficiency will be significantly promoted. However, the overall innovation efficiency of GEM-listed companies in the Chengyu city cluster is declining. Compared with the other two regions, the resources, infrastructure, and market environment in this region are relatively imperfect. However, it is not clear whether the innovation efficiency of GEM-listed companies in the Chengyu city cluster cannot be effectively improved owing to the macro-environment; this needs further research. Studying these companies by industry shows that there is obvious industry heterogeneity in GEM-listed companies in the YREB, and that there is a large gap in innovation efficiency among companies in different industries. To reduce this gap, we must make full use of innovation resources. Regarding the convergence trend, there is a significant *β* convergence trend in all three regions; in other words, they will eventually tend to a stable state. Improving this steady-state level as much as possible requires that each region achieve targeted improvement according to its own influencing factors.

The research conclusion of this paper is similar to the research result of Cheng Guangbin [52] to a certain extent; that is, there are significant differences in innovation efficiency between different regions, and there is a significant catch-up effect in regions with low innovation efficiency. However, the difference between this study and the previous research conclusion is that the measured regional innovation efficiency is different, and the delta innovation efficiency with the advantages of resources and geographical location is the lowest. The reason is that this study divides all GEM-listed companies into three regions. There are a large number of companies in the delta region. Due to the uneven level of

innovation efficiency, the final average result is affected by the extreme value, which lowers the overall innovation efficiency value.

*5.2. Conclusions*

Based on the panel data of GEM-listed companies in 11 provinces and cities of the YREB, this study examines their innovation efficiency and convergence trend, and draws the following conclusions: (1) From 2009 to 2021, the number of GEM-listed companies in the YREB increased substantially and was concentrated in the Yangtze River Delta region. These companies are distributed across 29 industries, and the industry agglomeration effect is obvious. Most GEM-listed companies belong to the industrial and public utilities industries. (2) There is room for improvement in the innovation efficiency of GEM-listed companies in China's YREB. The average value of innovation efficiency is ranked as: middle reaches of the Yangtze River > Chengyu city cluster > Yangtze River Delta region. The Malmquist index and its decomposition value of the three major regions show that the average Malmquist index is the Yangtze River Delta region > the provinces and cities in the middle reaches of the Yangtze River > Chengyu city cluster. (3) There is no $\sigma$ convergence trend in the YREB as a whole and its subregions. However, there are obvious absolute and conditional $\beta$-convergence trends. Moreover, exogenous factors have different effects on the improvement of innovation efficiency in different regions owing to differences in geographical location, economic development, and the factor endowments of different regions. (4) Innovation efficiency is influenced by many factors. Overall, the economic scale and government R&D expenditure have positive effects on the innovation efficiency of GEM-listed companies, while the scale of listed companies has negative effects on the innovation efficiency of GEM-listed companies in a certain range. Foreign investment has no significant effect on the innovation efficiency of GEM-listed companies.

*5.3. Policy Advice*

Based on the above empirical conclusions, this study proposes the following policy suggestions: First, adjust measures to local conditions and implement differentiation and gradient management according to the innovation situation in different regions. From the development trend of innovation efficiency in each region, the indicator value of technological progress is the highest in the Yangtze River Delta region, followed by the middle reaches of the Yangtze River, and the indicator value of the Chengyu city cluster is the lowest. This shows that the Yangtze River Delta region has the fastest development due to its resources and geographical advantages in the innovation of new technologies, and the Chengyu city cluster is obviously lagging behind in the exploration of new technologies. Therefore, the Chengyu city cluster and the middle reaches of the Yangtze River can attract some high-end talents to help these regions improve their technological innovation by issuing talent introduction and welfare policies, or introduce foreign investment to inject new energy into these regions to promote their technological innovation. For the Yangtze River Delta region, due to the rapid development of companies in the region, the government's support is ineffective due to the asymmetry of information, and government support even has a negative impact. This requires the government to follow up the development status quo of these enterprises, and relevant departments such as the Small and Medium-sized Enterprise Management Bureau update the information of the enterprise development stage in a timely manner to help the government provide support more accurately. Second, we should pay attention to the development gap between industries and reduce the unbalanced development among industries. From the innovation of different industries, we can see that most of the traditional industries, such as agriculture and manufacturing, show low innovation efficiency, while the entertainment industry and business service-related industries show high innovation efficiency. From this, we can see that if we want to achieve a balanced development among industries, the government should implement differentiated management, issue some subsidies and preferential policies and appropriate resource orientation to industries with low innovation efficiency to support the development of these

industries, while industries with high innovation efficiency should continue to maintain their existing development status. Third, maintain the coordinated development among regions. From the results of the convergence tests of each region, it can be seen that the convergence and steady-state rates of different regions are different. The Chengyu city cluster has the slowest convergence, and the other two regions are relatively fast. Therefore, the government should pay more attention to the development progress of the Chengyu city cluster, and encourage enterprises in the region to increase innovation investment through certain preferential loan policies. For the middle reaches of the Yangtze River region, the government should continue to increase its support for enterprise research and development, while the delta region can maintain its innovation efficiency growth by introducing foreign direct investment and increasing infrastructure.

Although this paper takes the Yangtze River Economic Zone as the research area, the policy suggestions put forward in the text can also be used for reference to improve the innovation efficiency of other regions to a certain extent. For example, the Yellow River Basin, as one of the important development areas in China, also deserves attention for its innovation efficiency. In fact, there are also spatial differences in the innovation efficiency of this region [53]. It also requires the government to adjust measures to local conditions, conduct differentiated management, and continuously narrow the regional differences through complementary resources. From the national perspective, the development of the industry is also uneven. The state should also attach great importance to it and help industries with low innovation efficiency actively to develop and narrow the gap between industries through macro allocation of resources and active guidance.

However, this study has some limitations. First, in the time range of research, some data in 2020 and 2021 were not yet released owing to the limited availability of data. Therefore, it is not possible to ascertain the innovation efficiency of GEM-listed companies in the YREB in the last two years. Second, there are fewer input and output indicators of innovation efficiency, which can be further enriched in the future. Third, when studying industry heterogeneity, some industries contain fewer companies. Future research needs to ascertain whether the innovation efficiency of these companies can represent the innovation efficiency of the entire industry. In the future, we can expand the research scope and number of companies in each industry to explore the heterogeneity of innovation efficiency among industries further.

**Author Contributions:** Conceptualization, Y.H., M.H. (Minghui Hua) and M.H. (Malan Huang); methodology, Y.H., M.H. (Minghui Hua) and S.W.; software, Y.H. and J.L.; validation, Y.H., M.H. (Minghui Hua) and M.H. (Malan Huang); formal analysis, M.H. (Minghui Hua), S.W. and M.H. (Malan Huang); investigation, Y.H., M.H. (Minghui Hua) and M.H. (Malan Huang); resources, Y.H.; data curation, M.H. (Malan Huang) and J.L.; writing—original draft preparation, Y.H. and M.H. (Minghui Hua); writing—review and editing, M.H. (Minghui Hua), M.H. (Malan Huang) and J.L.; visualization, M.H. (Malan Huang) and S.W.; supervision, Y.H., M.H. (Minghui Hua) and M.H. (Malan Huang); project administration, Y.H. and M.H. (Minghui Hua); funding acquisition, Y.H. All authors have read and agreed to the published version of the manuscript.

**Funding:** This research was funded by the National Key R&D Program of China (2021YFF0601005), Major National Social Science Projects (19ZDA085), China Postdoctoral Science Foundation (grant number: 2020M672317), and the National Natural Science Foundation Youth Project (grant number: 71904045).

**Institutional Review Board Statement:** Not applicable.

**Informed Consent Statement:** Not applicable.

**Data Availability Statement:** The data presented in this study are available upon reasonable request from the corresponding author.

**Acknowledgments:** We would like to thank all study participants for their help in the writing process of this paper.

**Conflicts of Interest:** The authors declare no conflict of interest.

## Appendix A

**Table A1.** Innovation efficiency values of 142 GEM listed companies in the Yangtze River Economic Belt from 2015 to 2019.

| Company Code | 2015 | 2016 | 2017 | 2018 | 2019 |
|---|---|---|---|---|---|
| 300006 | 0.872 | 1 | 0.634 | 1 | 1 |
| 300194 | 0.615 | 0.674 | 0.517 | 1 | 0.742 |
| 300275 | 1 | 1 | 0.844 | 0.694 | 0.796 |
| 300363 | 0.861 | 0.902 | 0.496 | 0.661 | 0.509 |
| 300019 | 0.922 | 0.672 | 0.805 | 0.946 | 0.882 |
| 300028 | 1 | 0.996 | 1 | 1 | 1 |
| 300092 | 1 | 0.827 | 1 | 0.987 | 0.955 |
| 300101 | 0.451 | 0.37 | 0.539 | 0.387 | 0.392 |
| 300249 | 0.76 | 0.689 | 0.732 | 0.764 | 0.68 |
| 300297 | 0.533 | 0.52 | 0.267 | 0.471 | 0.337 |
| 300366 | 0.848 | 0.633 | 0.513 | 0.577 | 0.49 |
| 300414 | 1 | 0.93 | 0.866 | 0.831 | 0.64 |
| 300425 | 0.993 | 0.962 | 1 | 1 | 1 |
| 300432 | 1 | 1 | 0.72 | 0.84 | 1 |
| 300434 | 1 | 1 | 0.937 | 1 | 1 |
| 300440 | 0.7 | 0.654 | 0.611 | 0.508 | 0.438 |
| 300463 | 0.879 | 0.845 | 0.555 | 1 | 0.663 |
| 300470 | 0.863 | 1 | 0.979 | 0.845 | 0.857 |
| 300471 | 0.795 | 1 | 1 | 0.832 | 0.78 |
| 300142 | 0.441 | 0.307 | 0.277 | 0.549 | 0.318 |
| 300288 | 0.624 | 0.672 | 0.842 | 1 | 0.93 |
| 300018 | 0.729 | 0.689 | 0.587 | 0.358 | 0.752 |
| 300041 | 0.862 | 0.923 | 0.632 | 0.724 | 0.756 |
| 300046 | 0.924 | 0.939 | 1 | 1.000- | 0.814 |
| 300054 | 0.93 | 0.897 | 0.764 | 0.749 | 0.852 |
| 300161 | 0.395 | 0.54 | 0.556 | 0.555 | 0.519 |
| 300184 | 1 | 1 | 1 | 1 | 1 |
| 300205 | 0.581 | 0.527 | 0.315 | 0.416 | 0.497 |
| 300220 | 0.845 | 0.964 | 1 | 1 | 1 |
| 300323 | 0.56 | 1 | 1 | 1 | 0.614 |
| 300387 | 0.906 | 1 | 0.994 | 0.952 | 1 |
| 300395 | 1 | 0.879 | 0.647 | 0.765 | 0.891 |
| 300035 | 1 | 1 | 0.966 | 0.927 | 0.889 |
| 300123 | 0.76 | 0.729 | 0.58 | 0.825 | 0.77 |
| 300187 | 0.955 | 0.969 | 0.704 | 0.497 | 0.948 |
| 300209 | 0.572 | 0.65 | 0.501 | 0.542 | 0.86 |
| 300298 | 0.972 | 0.78 | 0.761 | 0.636 | 0.647 |
| 300338 | 0.833 | 1 | 0.851 | 0.889 | 0.385 |
| 300345 | 0.906 | 0.94 | 1 | 1 | 1 |
| 300358 | 1 | 1 | 1 | 0.976 | 1 |
| 300433 | 0.691 | 0.677 | 0.586 | 0.542 | 0.366 |
| 300490 | 0.782 | 0.721 | 0.578 | 0.61 | 0.773 |
| 300066 | 1 | 0.912 | 0.643 | 0.886 | 0.99 |
| 300095 | 0.812 | 0.758 | 0.629 | 0.623 | 0.905 |
| 300294 | 1 | 1 | 1 | 1 | 0.827 |

**Table A1.** *Cont.*

| Company Code | 2015 | 2016 | 2017 | 2018 | 2019 |
|---|---|---|---|---|---|
| 300453 | 0.848 | 0.712 | 0.581 | 0.702 | 0.948 |
| 300472 | 0.824 | 1 | 0.558 | 1 | 0.921 |
| 300497 | 0.923 | 0.934 | 0.701 | 0.746 | 0.933 |
| 300013 | 0.66 | 0.623 | 0.499 | 0.584 | 0.585 |
| 300031 | 0.817 | 0.907 | 0.696 | 0.998 | 0.931 |
| 300091 | 0.55 | 0.617 | 0.663 | 0.729 | 0.69 |
| 300128 | 0.68 | 0.629 | 0.687 | 0.81 | 0.821 |
| 300141 | 0.863 | 0.666 | 0.594 | 0.788 | 0.852 |
| 300160 | 0.521 | 0.604 | 0.485 | 0.712 | 0.795 |
| 300165 | 0.424 | 0.571 | 0.578 | 0.547 | 0.598 |
| 300169 | 0.585 | 0.671 | 1 | 1 | 1 |
| 300172 | 0.915 | 0.73 | 0.656 | 0.801 | 0.729 |
| 300190 | 0.486 | 0.509 | 0.454 | 0.69 | 0.763 |
| 300196 | 0.479 | 0.57 | 0.566 | 0.636 | 0.675 |
| 300201 | 0.478 | 0.453 | 0.471 | 0.685 | 0.719 |
| 300215 | 0.514 | 0.514 | 0.417 | 0.574 | 0.542 |
| 300217 | 0.533 | 0.574 | 0.396 | 0.639 | 0.705 |
| 300228 | 0.46 | 0.555 | 0.715 | 0.776 | 0.622 |
| 300260 | 0.638 | 0.755 | 0.632 | 0.716 | 0.709 |
| 300261 | 0.331 | 0.375 | 0.297 | 0.494 | 0.553 |
| 300265 | 0.462 | 0.489 | 0.471 | 0.67 | 0.707 |
| 300279 | 0.615 | 0.554 | 0.493 | 0.534 | 0.717 |
| 300280 | 0.869 | 1 | 0.99 | 1 | 1 |
| 300284 | 0.632 | 0.498 | 0.523 | 0.551 | 0.63 |
| 300292 | 0.63 | 0.428 | 0.369 | 0.399 | 0.7 |
| 300304 | 0.656 | 0.658 | 0.506 | 0.654 | 0.57 |
| 300305 | 0.745 | 0.868 | 0.917 | 1 | 0.806 |
| 300320 | 0.596 | 0.684 | 0.499 | 0.646 | 0.686 |
| 300331 | 0.348 | 0.5 | 0.509 | 0.501 | 0.596 |
| 300337 | 0.479 | 0.765 | 0.882 | 0.837 | 0.791 |
| 300339 | 0.248 | 0.24 | 0.179 | 0.203 | 0.263 |
| 300342 | 0.724 | 0.723 | 0.537 | 0.587 | 0.517 |
| 300346 | 0.576 | 0.83 | 0.677 | 0.707 | 0.605 |
| 300382 | 0.772 | 0.815 | 0.75 | 0.833 | 0.795 |
| 300385 | 0.816 | 0.779 | 0.579 | 0.828 | 0.748 |
| 300390 | 0.831 | 0.685 | 0.564 | 0.807 | 0.783 |
| 300393 | 0.783 | 0.679 | 0.491 | 0.548 | 0.696 |
| 300394 | 0.856 | 0.716 | 0.632 | 0.7 | 0.662 |
| 300402 | 0.657 | 0.627 | 0.65 | 0.844 | 0.775 |
| 300416 | 0.654 | 0.743 | 0.539 | 0.666 | 0.72 |
| 300420 | 0.888 | 0.65 | 0.484 | 0.587 | 0.629 |
| 300421 | 0.769 | 0.846 | 0.838 | 0.758 | 0.77 |
| 300429 | 0.664 | 0.742 | 0.504 | 0.645 | 0.617 |
| 300447 | 0.7 | 0.535 | 0.47 | 0.5 | 0.562 |
| 300450 | 0.607 | 0.693 | 0.718 | 0.457 | 0.714 |
| 300466 | 0.867 | 0.755 | 0.501 | 0.574 | 0.631 |
| 300020 | 0.311 | 0.507 | 0.324 | 0.539 | 0.559 |
| 300025 | 0.407 | 0.392 | 0.299 | 0.623 | 0.699 |
| 300027 | 0.8 | 0.845 | 1 | 1 | 1 |
| 300032 | 0.515 | 0.677 | 0.366 | 0.391 | 0.744 |
| 300068 | 0.494 | 0.605 | 0.744 | 0.98 | 0.845 |
| 300076 | 0.813 | 0.681 | 0.688 | 1 | 1 |
| 300078 | 0.578 | 0.38 | 0.605 | 0.403 | 0.434 |
| 300100 | 0.644 | 0.593 | 0.508 | 0.551 | 0.58 |
| 300113 | 0.217 | 0.293 | 0.247 | 0.273 | 0.316 |
| 300118 | 0.472 | 0.562 | 0.542 | 0.503 | 0.529 |
| 300145 | 0.777 | 0.495 | 0.378 | 0.483 | 0.606 |

**Table A1.** *Cont.*

| Company Code | 2015 | 2016 | 2017 | 2018 | 2019 |
|---|---|---|---|---|---|
| 300203 | 0.284 | 0.291 | 0.238 | 0.292 | 0.336 |
| 300234 | 0.616 | 0.635 | 0.601 | 0.915 | 0.849 |
| 300244 | 0.406 | 0.566 | 0.48 | 0.654 | 0.92 |
| 300250 | 0.45 | 0.386 | 0.283 | 0.372 | 0.486 |
| 300266 | 0.575 | 0.538 | 0.486 | 0.505 | 0.677 |
| 300270 | 0.434 | 0.505 | 0.455 | 0.617 | 0.711 |
| 300283 | 0.527 | 0.62 | 0.567 | 0.754 | 0.844 |
| 300306 | 0.588 | 0.709 | 0.379 | 0.412 | 0.589 |
| 300307 | 0.444 | 0.388 | 0.423 | 0.737 | 0.603 |
| 300314 | 0.896 | 0.879 | 0.739 | 0.772 | 0.755 |
| 300316 | 0.401 | 0.429 | 0.492 | 0.428 | 0.532 |
| 300349 | 0.478 | 0.432 | 0.533 | 0.591 | 0.494 |
| 300351 | 0.751 | 0.574 | 0.504 | 0.6 | 0.501 |
| 300360 | 0.511 | 0.565 | 0.432 | 0.641 | 0.637 |
| 300411 | 0.944 | 0.954 | 0.669 | 0.388 | 0.651 |
| 300412 | 1 | 0.866 | 0.841 | 1 | 0.748 |
| 300435 | 0.848 | 0.899 | 0.915 | 0.858 | 0.871 |
| 300439 | 0.582 | 0.526 | 0.478 | 0.812 | 0.656 |
| 300441 | 0.91 | 0.787 | 0.667 | 0.774 | 0.745 |
| 300461 | 1 | 0.861 | 0.805 | 0.721 | 0.737 |
| 300008 | 1 | 1 | 0.702 | 0.581 | 0.714 |
| 300074 | 0.242 | 0.482 | 0.432 | 0.679 | 0.746 |
| 300129 | 0.462 | 0.632 | 0.798 | 0.842 | 0.781 |
| 300153 | 0.53 | 0.58 | 0.978 | 0.968 | 0.877 |
| 300171 | 0.494 | 0.425 | 0.452 | 0.655 | 0.588 |
| 300222 | 0.348 | 0.356 | 0.257 | 0.395 | 0.411 |
| 300230 | 0.6 | 0.464 | 0.364 | 0.62 | 0.732 |
| 300272 | 0.659 | 0.599 | 0.367 | 0.861 | 0.776 |
| 300326 | 0.516 | 0.693 | 0.68 | 1 | 0.715 |
| 300483 | 0.987 | 1 | 0.959 | 0.975 | 0.802 |
| 300009 | 0.468 | 0.404 | 0.276 | 0.336 | 0.309 |
| 300087 | 0.582 | 0.632 | 0.482 | 0.559 | 0.69 |
| 300088 | 1 | 1 | 1 | 1 | 0.835 |
| 300134 | 0.353 | 0.307 | 0.219 | 0.435 | 0.917 |
| 300218 | 0.448 | 0.434 | 0.488 | 0.591 | 0.639 |
| 300247 | 0.684 | 0.786 | 1 | 1 | 0.701 |
| 300274 | 0.656 | 0.472 | 0.452 | 0.461 | 0.573 |
| 300388 | 0.719 | 0.634 | 0.781 | 1 | 1 |
| 300452 | 1 | 1 | 0.873 | 0.994 | 0.908 |
| 300475 | 1 | 1 | 1 | 1 | 0.94 |

# Appendix B

**Table A2.** MI index and its decomposition value of 142 GEM listed companies in the Yangtze River Economic Belt from 2015 to 2019.

| Company Code | Technical Efficiency | Technical Progress | Pure Technical Efficiency | Scale Efficiency | Malmquist Index |
|---|---|---|---|---|---|
| 300006 | 1.035 | 0.902 | 1 | 1.035 | 0.934 |
| 300194 | 1.048 | 0.769 | 1.036 | 1.012 | 0.806 |
| 300275 | 0.945 | 0.801 | 0.981 | 0.963 | 0.756 |
| 300363 | 0.877 | 0.883 | 0.945 | 0.927 | 0.774 |
| 300019 | 0.989 | 0.85 | 1.011 | 0.978 | 0.841 |
| 300028 | 1 | 1.27 | 1 | 1 | 1.27 |
| 300092 | 0.989 | 0.815 | 0.994 | 0.995 | 0.805 |

**Table A2.** *Cont.*

| Company Code | Technical Efficiency | Technical Progress | Pure Technical Efficiency | Scale Efficiency | Malmquist Index |
|---|---|---|---|---|---|
| 300101 | 0.966 | 0.868 | 0.952 | 1.014 | 0.838 |
| 300249 | 0.973 | 0.914 | 0.98 | 0.993 | 0.889 |
| 300297 | 0.892 | 0.892 | 0.947 | 0.942 | 0.796 |
| 300366 | 0.872 | 0.823 | 0.956 | 0.912 | 0.718 |
| 300414 | 0.895 | 0.789 | 0.968 | 0.924 | 0.706 |
| 300425 | 1.002 | 0.802 | 1.001 | 1.001 | 0.803 |
| 300432 | 1 | 0.903 | 1 | 1 | 0.903 |
| 300434 | 1 | 0.687 | 1 | 1 | 0.687 |
| 300440 | 0.889 | 0.833 | 0.956 | 0.931 | 0.741 |
| 300463 | 0.932 | 0.948 | 1 | 0.932 | 0.883 |
| 300470 | 0.998 | 0.803 | 1.005 | 0.993 | 0.801 |
| 300471 | 0.995 | 1.043 | 0.973 | 1.022 | 1.038 |
| 300142 | 0.921 | 1.159 | 1.01 | 0.913 | 1.068 |
| 300288 | 1.105 | 0.791 | 1.003 | 1.102 | 0.874 |
| 300018 | 1.008 | 1.071 | 1.017 | 0.991 | 1.08 |
| 300041 | 0.968 | 1.017 | 0.973 | 0.994 | 0.984 |
| 300046 | 0.969 | 1.123 | 0.971 | 0.998 | 1.088 |
| 300054 | 0.978 | 1.152 | 0.973 | 1.005 | 1.127 |
| 300161 | 1.071 | 1.051 | 1.047 | 1.023 | 1.125 |
| 300184 | 1 | 1.189 | 1 | 1 | 1.189 |
| 300205 | 0.962 | 1.203 | 1.033 | 0.931 | 1.157 |
| 300220 | 1.043 | 1.024 | 1 | 1.043 | 1.068 |
| 300323 | 1.023 | 1.078 | 0.946 | 1.081 | 1.104 |
| 300387 | 1.025 | 0.994 | 1.025 | 1 | 1.019 |
| 300395 | 0.971 | 1.107 | 0.99 | 0.982 | 1.076 |
| 300035 | 0.971 | 0.938 | 0.99 | 0.981 | 0.911 |
| 300123 | 1.003 | 1.148 | 1.036 | 0.968 | 1.152 |
| 300187 | 0.998 | 1.157 | 0.998 | 1 | 1.155 |
| 300209 | 1.107 | 1.131 | 1.089 | 1.017 | 1.253 |
| 300298 | 0.903 | 1.143 | 0.974 | 0.927 | 1.032 |
| 300338 | 0.825 | 1.006 | 0.856 | 0.964 | 0.83 |
| 300345 | 1.025 | 1.061 | 1.011 | 1.014 | 1.088 |
| 300358 | 1 | 1.051 | 1 | 1 | 1.051 |
| 300433 | 0.853 | 1.099 | 1 | 0.853 | 0.937 |
| 300490 | 0.997 | 1.098 | 1.018 | 0.979 | 1.095 |
| 300066 | 0.998 | 1.086 | 1 | 0.998 | 1.083 |
| 300095 | 1.028 | 1.066 | 1.027 | 1 | 1.096 |
| 300294 | 0.954 | 1.11 | 1 | 0.954 | 1.058 |
| 300453 | 1.028 | 1.187 | 1.009 | 1.019 | 1.221 |
| 300472 | 1.028 | 1.008 | 1.032 | 0.997 | 1.037 |
| 300497 | 1.003 | 1.13 | 1.019 | 0.984 | 1.133 |
| 300013 | 0.97 | 1.258 | 1.045 | 0.928 | 1.22 |
| 300031 | 1.033 | 1.162 | 1.034 | 0.999 | 1.2 |
| 300091 | 1.058 | 1.104 | 1.122 | 0.943 | 1.168 |
| 300128 | 1.048 | 1.045 | 0.997 | 1.052 | 1.095 |
| 300141 | 0.997 | 1.223 | 1.02 | 0.977 | 1.219 |
| 300160 | 1.111 | 1.128 | 1.127 | 0.986 | 1.253 |
| 300165 | 1.089 | 1.259 | 1.117 | 0.975 | 1.371 |
| 300169 | 1.144 | 1.073 | 1.072 | 1.067 | 1.227 |
| 300172 | 0.945 | 1.216 | 0.992 | 0.952 | 1.148 |
| 300190 | 1.119 | 1.125 | 1.131 | 0.989 | 1.26 |
| 300196 | 1.089 | 1.051 | 1.076 | 1.013 | 1.145 |
| 300201 | 1.108 | 1.198 | 1.122 | 0.987 | 1.327 |
| 300215 | 1.014 | 1.303 | 1.093 | 0.928 | 1.32 |
| 300217 | 1.072 | 1.132 | 1.098 | 0.977 | 1.214 |
| 300228 | 1.079 | 1.024 | 1.023 | 1.054 | 1.104 |
| 300260 | 1.027 | 1.187 | 1.087 | 0.945 | 1.219 |

**Table A2.** *Cont.*

| Company Code | Technical Efficiency | Technical Progress | Pure Technical Efficiency | Scale Efficiency | Malmquist Index |
|---|---|---|---|---|---|
| 300261 | 1.137 | 1.148 | 1.16 | 0.981 | 1.305 |
| 300265 | 1.112 | 1.072 | 1.113 | 1 | 1.193 |
| 300279 | 1.039 | 1.036 | 1.081 | 0.961 | 1.076 |
| 300280 | 1.036 | 1.256 | 1.035 | 1.001 | 1.301 |
| 300284 | 0.999 | 1.085 | 0.995 | 1.004 | 1.085 |
| 300292 | 1.027 | 1.012 | 1.031 | 0.996 | 1.039 |
| 300304 | 0.966 | 1.3 | 1.03 | 0.937 | 1.255 |
| 300305 | 1.02 | 1.182 | 1.04 | 0.981 | 1.206 |
| 300320 | 1.036 | 1.102 | 1.092 | 0.948 | 1.141 |
| 300331 | 1.144 | 1.152 | 1.221 | 0.937 | 1.318 |
| 300337 | 1.134 | 0.842 | 1.056 | 1.074 | 0.954 |
| 300339 | 1.015 | 1.175 | 1.095 | 0.926 | 1.192 |
| 300342 | 0.919 | 1.313 | 1.035 | 0.888 | 1.207 |
| 300346 | 1.012 | 1.193 | 1.06 | 0.955 | 1.208 |
| 300382 | 1.007 | 1.325 | 1.026 | 0.982 | 1.335 |
| 300385 | 0.978 | 1.238 | 1.022 | 0.957 | 1.211 |
| 300390 | 0.985 | 1.271 | 1.022 | 0.964 | 1.252 |
| 300393 | 0.971 | 0.976 | 1.005 | 0.966 | 0.948 |
| 300394 | 0.938 | 1.327 | 1.005 | 0.933 | 1.245 |
| 300402 | 1.042 | 1.259 | 1.077 | 0.968 | 1.312 |
| 300416 | 1.025 | 1.318 | 1.076 | 0.952 | 1.351 |
| 300420 | 0.918 | 1.212 | 1.014 | 0.905 | 1.112 |
| 300421 | 1 | 1.201 | 1.031 | 0.97 | 1.201 |
| 300429 | 0.982 | 1.283 | 1.058 | 0.928 | 1.26 |
| 300447 | 0.946 | 1.262 | 1.052 | 0.9 | 1.194 |
| 300450 | 1.041 | 1.176 | 1.085 | 0.96 | 1.225 |
| 300466 | 0.924 | 1.324 | 0.994 | 0.929 | 1.223 |
| 300020 | 1.158 | 1.059 | 1.114 | 1.04 | 1.226 |
| 300025 | 1.145 | 1.162 | 1.166 | 0.982 | 1.33 |
| 300027 | 1.057 | 1.544 | 1 | 1.057 | 1.633 |
| 300032 | 1.096 | 0.996 | 1.02 | 1.075 | 1.092 |
| 300068 | 1.144 | 0.913 | 1 | 1.144 | 1.044 |
| 300076 | 1.053 | 1.372 | 1.05 | 1.003 | 1.445 |
| 300078 | 0.931 | 1.22 | 1.03 | 0.904 | 1.135 |
| 300100 | 0.974 | 1.019 | 0.942 | 1.035 | 0.992 |
| 300113 | 1.099 | 1.254 | 1.19 | 0.924 | 1.379 |
| 300118 | 1.029 | 0.914 | 1 | 1.029 | 0.941 |
| 300145 | 0.94 | 1.08 | 0.973 | 0.966 | 1.015 |
| 300203 | 1.043 | 1.002 | 1.047 | 0.996 | 1.046 |
| 300234 | 1.083 | 1.276 | 1.075 | 1.008 | 1.382 |
| 300244 | 1.227 | 0.975 | 1.138 | 1.078 | 1.196 |
| 300250 | 1.019 | 1.333 | 1.132 | 0.901 | 1.359 |
| 300266 | 1.042 | 1.105 | 1.071 | 0.973 | 1.152 |
| 300270 | 1.131 | 1.342 | 1.122 | 1.008 | 1.518 |
| 300283 | 1.125 | 1.162 | 1.126 | 0.999 | 1.307 |
| 300306 | 1 | 1.3 | 1.084 | 0.923 | 1.301 |
| 300307 | 1.079 | 1.169 | 1.036 | 1.042 | 1.261 |
| 300314 | 0.958 | 1.308 | 1.013 | 0.946 | 1.253 |
| 300316 | 1.074 | 1.144 | 1.19 | 0.903 | 1.229 |
| 300349 | 1.009 | 1.284 | 1.143 | 0.882 | 1.295 |
| 300351 | 0.904 | 1.326 | 0.968 | 0.933 | 1.198 |
| 300360 | 1.056 | 1.264 | 1.096 | 0.964 | 1.336 |
| 300411 | 0.911 | 1.147 | 0.996 | 0.915 | 1.045 |
| 300412 | 0.93 | 1.377 | 1 | 0.93 | 1.28 |
| 300435 | 1.007 | 1.265 | 1.012 | 0.994 | 1.273 |
| 300439 | 1.03 | 1.102 | 0.976 | 1.055 | 1.135 |

**Table A2.** *Cont.*

| Company Code | Technical Efficiency | Technical Progress | Pure Technical Efficiency | Scale Efficiency | Malmquist Index |
|---|---|---|---|---|---|
| 300441 | 0.951 | 1.284 | 1.016 | 0.936 | 1.221 |
| 300461 | 0.927 | 1.309 | 0.973 | 0.953 | 1.213 |
| 300008 | 0.919 | 0.939 | 0.967 | 0.951 | 0.863 |
| 300074 | 1.325 | 1.319 | 1.277 | 1.038 | 1.747 |
| 300129 | 1.14 | 0.949 | 1.017 | 1.121 | 1.082 |
| 300153 | 1.134 | 1.073 | 1.13 | 1.004 | 1.217 |
| 300171 | 1.044 | 1.184 | 1.031 | 1.013 | 1.237 |
| 300222 | 1.043 | 1.191 | 0.994 | 1.049 | 1.242 |
| 300230 | 1.051 | 1.125 | 1.09 | 0.964 | 1.182 |
| 300272 | 1.042 | 1.3 | 1.076 | 0.968 | 1.355 |
| 300326 | 1.085 | 1.194 | 1.035 | 1.048 | 1.295 |
| 300483 | 0.949 | 1.29 | 1 | 0.949 | 1.225 |
| 300009 | 0.901 | 1.251 | 1.098 | 0.821 | 1.128 |
| 300087 | 1.043 | 1.173 | 1.087 | 0.96 | 1.223 |
| 300088 | 0.956 | 1.001 | 1 | 0.956 | 0.957 |
| 300134 | 1.27 | 1.145 | 1.142 | 1.112 | 1.453 |
| 300218 | 1.093 | 1.136 | 1.112 | 0.982 | 1.242 |
| 300247 | 1.006 | 1.274 | 1.01 | 0.996 | 1.282 |
| 300274 | 0.967 | 1.018 | 1 | 0.967 | 0.984 |
| 300388 | 1.086 | 0.985 | 1.056 | 1.029 | 1.07 |
| 300452 | 0.976 | 1.295 | 0.99 | 0.986 | 1.264 |
| 300475 | 0.985 | 1.084 | 0.996 | 0.989 | 1.068 |

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
