# Peer review of "Dynamic Transition and Convergence Trend of the Innovation Efficiency among Companies Listed on the Growth Enterprise Market in the Yangtze River Economic Belt—Empirical Analysis Based on DEA—Malmquist Model"

_sustainability, doi:10.3390/su14095269_

Round 1

Reviewer 1 Report

The article proposed for review is devoted to the study of the innovative efficiency of companies placed on the Growth Enterprise Market (GEM) in the Yangtze River Economic Belt (YREB) using a combination of advanced economic and mathematical methods (DEA-Malmquist model, σ-convergence, and β-convergence models). The article's topic is relevant for economic science because the efficiency of production activity of real companies at the micro-level significantly affects the competitiveness of national economies. Despite the overall positive impression of the article, the lack of substantiation of the connection between the article's topic and the goals of "Sustainability" is critical. The empirical analysis of GEM-listed companies is valuable, but the sustainability aspect of YREB provinces is uncovered. Improving the innovative efficiency of companies may be a goal in the medium term, but in the strategic perspective, we need to take into account such factors as ecology, social responsibility, people's satisfaction with their life, and others. 

We suggest that the authors pay attention to the following reserves to improve the article:

1. Connect the study to the aspects of sustainable development and think about how the conclusions obtained in the article can be applied to other regions of China or other countries.
2. The sources related to applying mathematical methods to evaluate innovation efficiency are reviewed in detail; however, there is no review of the sources that researched YREB companies.
3. The introduction should also reflect the authors' initial hypotheses, which are confirmed or refuted from empirical analysis.
4. It is necessary to justify the research design and the choice of this particular combination of methods for the empirical analysis.
5. It is desirable to compare the growth of the number of companies by year and their innovative efficiency with external factors and the economic policy of regional governments, which probably played not the least role in these processes.
6. It is necessary to give extended comments to Fig. 2, which reflects Malmquist index (MI) values by year. In 2017-2018, the (MI) values became very high due to an increase in the "Advances in technology" component for the middle reaches of the Yangtze River and Yangtze River Delta region and low for the Chengyu city cluster. What factors could be responsible for these changes?
7. It would be useful to give a more extended interpretation of the coefficient values in Table 10 because there are some anomalies here. For example, government support positively affects the innovative efficiency of companies in the Middle reaches of the Yangtze River and negatively affects the innovative efficiency of companies in the Yangtze River delta. 
8. Not all abbreviations in the text are deciphered; it is desirable to indicate the decipherment of abbreviations DMU, ST, ST* and check the decoding of other abbreviations.
9. Also, it is necessary to check lines 158-160, 185-186, and 237, which have inaccuracies.

Author Response

We modified the comments made by the review experts in the text. There are three documents in total. The first document is a manuscript containing any modification marks, the second document is a revised manuscript without modification trace but with marks, and the third document is a reply to the comments made by the experts.

Reviewer 2 Report

The topic is very interesting, and the research method is quite comprehensive. But you should improve the writing style and references.

Please improve your writing style for a few sentences in the abstract (re-phrase):

The Yangtze River Economic Belt(YREB) occupies an important economic 11 position in China and has great research significance. (the second part of the sentence should be re-phrased)

The innovation efficiency (of) these companies 21 has a lot of room for improvement.

Wadud and White (2000)[1] and 61 other scholars use the SFA and DEA methods *please indicate a few sources of other scholars (61-62 lines)

The same is for Leejy 63 (2005)[2] and other scholars (63-64).

Thus, whenever you state „and other related studies” you should indicate at least 2-3 sources.

There are some errors related to some references that you need to solve. This is a case in many places trough the manuscript (if there is a technical issue with this, please discuss it with the editor):

[Error! 65 Reference source not found.], Fang (2020)[Error! Reference source not found.], Ming et 66 al.(2019)[Error! Reference source not found.], and Iglesias, Castellanos, and Seijas 67 (2010)[Error! Reference source not found.]

(Grammar issue) On (In) the research sample, there is a plethora of literature investigating innovation efficiency in the macro region and industry.

What is the meaning of this sentence: Moreover, only a few companies have been published (125) what has been published (companies or data about them?) = re-phrase it

The Malmquist index, proposed by Malmquist (you could indicate the source here, otherwise exclude the text proposed by….)

(237) Jiangsu Province had the largest number of GEM- 236 listed companies: 152 135 and 35 from Zhejiang Province. What is 152 135 and 35 (comma is missing)?

Irrational missing data processing. To ensure data integrity, this study first deleted company samples with a large amount of missing input-output index data. For patent application data, this study used the corresponding data in the CNRDS database to match. The rest of the input-output index data were obtained from the CSMAR database (Do CNRD and CSMAR databases use the same methodology since you combined data from both sources? Please elaborate in this part)

481 Regarding foreign direct investment, on the one hand, (why repeat twice FDI in the same sentence?) foreign direct investment can increase 482 market vitality,

The conclusion usually came after discussion! And the discussion should reflect the differences in your findings compared to other authors. I personally see your discussion as an extension of your conclusion, but please think about this.

Author Response

(The authors gave the same response as above.)

Reviewer 3 Report

There is no hypothesis in the article while the results allow hypotheses to be made.

The discussion about "innovation efficiency" is not concluded - in the paper we have the input data (Appendices) but we do not know how the input data are calculated. The authors should consider to refer to the article by Cruz-Cazares et al. 2013 (https://doi.org/10.1016/j.respol.2013.03.012 ) where the same method is used but the innovation efficiency is calculated based on innovation management performance.

DEA-Malmquist is performed well and according to the literature - the limitation of the method are clearly presented.

The results show the role of Chengyu city cluster which has the value of "knowledge area" for the China's development.

The first reference (line 627,628): Outline of the Development Plan of the Yangtze River Economic Belt is not available on-line. The link is not active.

The policy advice is very general - I don't find it useful for policy makers.

Author Response

Thank you very much for your guidance to this article. We have prepared three documents. The first document is a reply to your suggestion, the second document is a document with any modification marks, and the third document is a document without modification trace but marked.

Round 2

Reviewer 1 Report

It can be stated that the authors of the article have done much work to improve it. At the moment, the article is almost ready for publication. Nevertheless, it is necessary to draw the attention of the authors of the article to the correction of some points.

1. The article still lacks a review of papers of other authors involved in researching the different aspects of the Yangtze River Economic Belt economy. It is recommended to examine the following works:

Cheng, Y., Liu, W., & Lu, J. (2017). Financing innovation in the Yangtze River Economic Belt: rationale and impact on firm growth and foreign trade. Canadian Public Policy43(S2), S122-S135.

Liu, Y., Zhang, X., Pan, X., Ma, X., & Tang, M. (2020). The spatial integration and coordinated industrial development of urban agglomerations in the Yangtze River Economic Belt, China. Cities, 104, 102801.

Zou, L., Cao, X. Z., & Zhu, Y. W. (2021). Research on regional high-tech innovation efficiency and influence factors: evidence from Yangtze river economic belt in China. Complexity, 2021.

Meng, D., Wei, G., & Sun, P. (2020). Analyzing the Characteristics and Causes of Location Spatial Agglomeration of Listed Companies: An Empirical Study of China's Yangtze River Economic Belt. Complexity2020.

Yi, M., Wang, Y., Yan, M., Fu, L., & Zhang, Y. (2020). Government R&D subsidies, environmental regulations, and their effect on green innovation efficiency of manufacturing industry: Evidence from the Yangtze River economic belt of China. International Journal of Environmental Research and Public Health17(4), 1330.

Li, X., Lu, Y., & Huang, R. (2021). Whether foreign direct investment can promote high-quality economic development under environmental regulation: evidence from the Yangtze River Economic Belt, China. Environmental Science and Pollution Research28(17), 21674-21683.

2. It is desirable to move table 11 between lines 611 and 612 and add in this table the decoding of X1, X2, X3, and X4.

3. It is necessary to eliminate double spaces on lines 92, 116, and 205.

Author Response

(The authors gave the same response as above.)
